# Significant wintertime PM$_{2.5}$ mitigation in the Yangtze River Delta, China from 2016 to 2019: observational constraints on anthropogenic emission controls

Liqiang Wang[1], Shaocai Yu[*,1,2], Pengfei Li[*,3,1], Xue Chen[1], Zhen Li[1], Yibo Zhang[1], Mengying Li[1], Khalid Mehmood[1], Weiping Liu[1], Tianfeng Chai[4], Yannian Zhu[5], Daniel Rosenfeld[6], and John H. Seinfeld[2]

[1]Research Center for Air Pollution and Health; Key Laboratory of Environmental Remediation and Ecological Health, Ministry of Education, College of Environment and Resource Sciences, Zhejiang University, Hangzhou, Zhejiang 310058, P.R. China
[2]Division of Chemistry and Chemical Engineering, California Institute of Technology, Pasadena, CA 91125, USA
[3]College of Science and Technology, Hebei Agricultural University, Baoding, Hebei 071000, P.R. China
[4]Air Resources Laboratory, NOAA, Cooperative Institute for Satellite Earth System Studies (CISESS), University of Maryland, College Park, USA
[5]Meteorological Institute of Shananxi Province, 36 Beiguanzhengjie, Xi'an 710015, China
[6]Institute of Earth Science, The Hebrew University of Jerusalem, Jerusalem, Israel

Correspondence to: Shaocai Yu (shaocaiyu@zju.edu.cn); Pengfei Li (lpf_zju@163.com)

**To be submitted to**

**Atmospheric Chemistry and Physics**

**ABSTRACT**

Ambient fine particulate matter ($PM_{2.5}$) mitigation relies strongly on anthropogenic emission control measures, the actual effectiveness of which is challenging to pinpoint owing to the complex synergies between anthropogenic emissions and meteorology. Here, observational constraints on model simulations allow us to derive not only reliable $PM_{2.5}$ evolution but also accurate meteorological fields. On this basis, we isolate meteorological factors to achieve reliable estimates of surface $PM_{2.5}$ responses to both long-term and emergency emission control measures from 2016 to 2019 over the Yangtze River Delta (YRD), China. The results show that long-term emission control strategies play a crucial role in curbing $PM_{2.5}$ levels, especially in the megacities and other areas with abundant anthropogenic emissions. The G20 summit hosted in Hangzhou in 2016 provides a unique and ideal opportunity involving the most stringent, even unsustainable, emergency emission control measures. These emergency measures lead to the largest decrease (~ 35 μg/m$^3$, ~ 59%) in $PM_{2.5}$ concentrations in Hangzhou. The hotspots also emerge in megacities, especially in Shanghai (32 μg/m$^3$, 51%), Nanjing (27 μg/m$^3$, 55%), and Hefei (24 μg/m$^3$, 44%) because of the emergency measures. Compared to the long-term policies from 2016 to 2019, the emergency emission control measures implemented during the G20 Summit achieve more significant decreases in $PM_{2.5}$ concentrations (17 μg/m$^3$ and 41%) over most of the whole domain, especially in Hangzhou (24 μg/m$^3$, 48%) and Shanghai (21 μg/m$^3$, 45%). By extrapolation, we derive insight into the magnitude and spatial distribution of $PM_{2.5}$ mitigation potential across the YRD, revealing significantly additional room for curbing $PM_{2.5}$ levels.

## 1 INTRODUCTION

Anthropogenic induced fine particulate matter (particulate matter with an aerodynamic diameter smaller than 2.5 μm, hereinafter denoted as $PM_{2.5}$) is a principal object of air pollution control in China (Huang et al., 2014; Zhang et al., 2015). Moreover, the government has made major strides in curbing anthropogenic emissions (e.g., $SO_2$, $NO_x$, and CO) via both long-term and emergency measures during the past decade (Yan et al., 2018; Yang et al., 2019; Zhang et al., 2012). However, owing to the complex synergy of chemistry and meteorology (Seinfeld and Pandis, 2016), the extent to which these measures have abated $PM_{2.5}$ pollution, as well as the attainable mitigation potential, remains unclear (An et al., 2019).

The main challenge involves reliably representing substantial and rapid changes in anthropogenic emissions resulting from both long-term and emergency control measures (Chen et al., 2019; Cheng et al., 2019; Zhang et al., 2014; Yang et al., 2016; Zhai et al., 2019; Zhang et al., 2019; Zhong et al., 2018). To gain timely insight into variations in anthropogenic emissions, considerable efforts went into establishing detailed bottom-up emissions and derived valuable findings (Cheng et al., 2019; Zhang et al., 2019). Yet bottom-up inventories were built on the basis of activity data as well as emission factors. These input data can be absent or outdated, likely leading to misunderstandings of anthropogenic impacts, particularly in terms of the

magnitude (Jiang et al., 2018). Recent studies applied available observations to construct multilinear regression models (emission-based or meteorology-related), thus allowing us to separate contributions from anthropogenic emissions and meteorology to some extent (Zhai et al., 2019; Zhong et al., 2018). However, the uncertainties in bottom-up inventories and meteorology remained. Here we switched to observational constraints on a state-of-the-art chemical model. This can be a potential way to tackle this challenge.

Since 2013, the China National Environmental Monitoring Center (CNEMC) has established 1415 ground-based $PM_{2.5}$ measurement sites across 367 key cities (Zhang and Cao, 2015). In contrast to satellite observations with sparse spatiotemporal coverages (Ma et al., 2014, 2015; Xue et al., 2019), these ground sites can provide hourly $PM_{2.5}$ concentrations at high spatial resolution in urban areas. Data assimilation (DA) methods that have been widely used in meteorology can be extended to integrate those continuous observational constraints with chemical transport models (CTMs) (Bocquet et al., 2015; Chai et al., 2017; Gao et al., 2017; Jung et al., 2019; Ma et al., 2019). It has been demonstrated that the capability of several representative DA methods, such as the optimal interpolation (OI) (Chai et al., 2017), 3D/4D variational methods (Li et al., 2016), and the ensemble Kalman filter algorithm (Chen et al., 2019), can bridge the estimation gaps between observed and simulated results. Thus, observational constraints can be taken full advantage of to identify the effects of anthropogenic emission controls.

From the perspective of policymaking, 2016 was a special year for air pollution control in China. Since 2013, the Chinese government instituted extensive policies, such as the Air Pollution Prevention and Control Action Plan. These strategies were initiated and implemented through generally shutting down or relocating high emission traditional industrial enterprises (Sheehan et al., 2014; Shi et al., 2016; Xie et al., 2015). Starting from January 1, 2016, the relevant law, as well as the "Blue Sky Battle Plan", came into full effect and profoundly shifted how China prioritized air quality management(Feng and Liao, 2016; Li et al., 2019c). Hence, we address the impact of long-term emission control strategies on $PM_{2.5}$ mitigation from 2016 onward.

The G20 summit hosted in Hangzhou in 2016 (hereinafter termed the G20 summit) provides a unique and ideal opportunity to further explore the attainable $PM_{2.5}$ mitigation potential across the Yangtze River Delta (YRD) (Li et al., 2017c; Ma et al., 2019; Shu et al., 2019; Yang et al., 2019). Prior to and during this period, the Chinese government enforced historically strictest, even unsustainable, emergency emission control measures, including significant control, even cessation, of factory operations, restrictions on vehicles in the region, thus achieving significant $PM_{2.5}$ abatement at specific locations (e.g., Hangzhou) (Ji et al., 2018; Li et al., 2017c; Yang et al., 2019). Those measures were conducted across the whole YRD (including Zhejiang province, Shanghai municipality, Jiangsu province, and Anhui province), particularly in Hangzhou that served as the host city (Li et al., 2019b, 2017c; Ni et al., 2020; Yu et al., 2018). Li et al. (2017) assumed that most of anthropogenic emissions (e.g., those from industry, power plant, residential, and on-road transportation sectors) were reduced by around 50%. The role of

these emergency emission control measures, that is, the relatively localized PM$_{2.5}$ mitigation potential, can thus be identified,
and further extended to the entire YRD.
To quantify the effectiveness of the emission control strategies, we constrained a state-of-the-art CTM by a reliable DA method
with extensive chemical and meteorological observations. This comprehensive technical design provides a crucial advance in
isolating the influences of emission changes and meteorological perturbations over the YRD from 2016 to 2019, thus deriving
estimates of PM$_{2.5}$ responses to both long-term and emergency emission control measures, and establishing the first map of
the PM$_{2.5}$ mitigation potential across the YRD.
**2 MATERIALS AND METHODS**
**2.1 The two-way coupled WRF-CMAQ model**
The two-way coupled Weather Research and Forecasting (WRF) and Community Multiscale Air Quality (CMAQ) model (the
WRF-CMAQ model), as the key core of the DA system, was applied to investigate the ambient PM$_{2.5}$ feedbacks under different
constraining circumstances (Byun and Schere, 2006; Wong et al., 2012; Yu et al., 2013). We utilized the CB05 and AERO6
modules for gas-phase chemistry and aerosol evolution (Carlton et al., 2010; Yarwood et al., 2005), respectively. Both
secondary inorganic and organic aerosol (i.e., SIA and SOA) were thus explicitly treated with the AERO6 scheme in the WRF-
CMAQ model. Together with the ISORROPIA II thermodynamic equilibrium module (Fountoukis and Nenes, 2007), SIA in
the Aitken and accumulation modes (Binkowski and Roselle, 2003) was assumed to be in thermodynamic equilibrium with
the gas phase, while that in the coarse mode was treated dynamically. SOA was formed via gas-, aqueous-, and aerosol-phase
oxidation processes, such as in-cloud oxidation of glyoxal and methylglyoxal, absorptive partitioning of condensable oxidation
of monoterpenes, long alkanes, low-yield aromatic products (based on m-xylene data), and high-yield aromatics, and NO$_x$-
dependent yields from aromatic compounds (Carlton et al., 2010). The subsequent reaction products can be divided into two
groups: non-volatile semi-volatile. Such treatments have been widely used and comprehensively validated. Longwave and
shortwave radiation were both treated using the RRTMG radiation scheme (Clough et al., 2005). Related land surface energy
balance and planetary boundary layer simulations were included in the Pleim-Xiu land surface scheme (Xiu and Pleim, 2001)
and the asymmetric convective model (Pleim, 2007b, 2007a), respectively. The two-moment Morrison cloud microphysics
scheme(Morrison and Gettelman, 2008) and the Kain-Fritsch cumulus cloud scheme (Kain, 2004) were employed for
simulating aerosol-cloud interactions and precipitation. Default settings in the model were used to prescribe chemical initial
and boundary conditions. A spin-up period of seven days was carried out in advance to eliminate artefacts associated with
initial conditions. Meteorological initial and boundary conditions were obtained from the ECMWF reanalysis dataset with the
spatial resolution of $1° \times 1°$ and temporal resolution of 6 hours (http://www.ecmwf.int/products/data, last access: 7 March
2020). Biogenic and dust emissions were calculated on-line using the Biogenic Emission Inventory System version 3.14
(BEISv3.14) (Carlton and Baker, 2011) and a windblown dust scheme embedded in CMAQ (Choi and Fernando, 2008),
respectively.
The horizontal domain of the model covered mainland China by a 395 $\times$ 345 grid with a 12 km horizontal resolution following
a Lambert Conformal Conic projection (Figure 1). In terms of the vertical configuration, 29 sigma-pressure layers ranged from
the surface to the upper level pressure of 100 hPa, 20 layers of which are located below around 3 km to derive finer
meteorological and chemical characteristics within the planetary boundary layer.
As a state-of-the-art CTM, the WRF-CMAQ model has been widely used to simulate spatiotemporal $PM_{2.5}$ distributions at
regional scales. However, model biases remain, mainly due to imperfect representations of chemical and meteorological
processes. Inaccurate anthropogenic emissions will exacerbate these biases. Therefore, external constraints on simulated results
enforced by the DA method will be taken into account in order to optimize spatiotemporal $PM_{2.5}$ distributions (Bocquet et al.,

125    2015).

**2.2 Anthropogenic emissions**
The anthropogenic emissions were obtained from the Multi-resolution Emission Inventory for China version 1.2 (MEIC)(Li
et al., 2017b), which contained primary species (e.g., primary $PM_{2.5}$, $SO_2$, $NO_x$, CO, and $NH_4$) from five anthropogenic sectors
(i.e., agriculture, power plant, industry, residential, and transportation). This inventory was initially designed with the spatial
resolution of 0.25 $°$ $\times$ 0.25 $°$ and thus needed to be reallocated to match the domain configuration (i.e., 12km $\times$ 12km) in the
study.
Recent findings show that MEIC generally provides reasonable estimates of total anthropogenic emissions for several typical
regions in China, such as the Beijing-Tianjin-Hebei region, the YRD, and the Pearl River Delta region (Li et al., 2017b).
Nevertheless, large uncertainties in spatial proxies (e.g., population density and road networks) still exist within these specific
regions (Geng et al., 2017). More, MEIC was originally constructed for the 2016 base year. Hence, owing to the impact of the
long-term emission control measures, MEIC was considered to be inappropriate for this study period (i.e., 2019).
Comparatively, emergency control measures could give rise to much more significant emission controls in the short term,
thereby leading to further uncertainties.
**2.3 Observational network**
To track real-time air quality in China, the National Environmental Monitoring Center (CNEMC, http://www.cnemc.cn/, last
access: 7 March 2020) has established 1415 sites across 367 cities since 2013 (Figure 1). Among these, 244 monitoring sites
were densely distributed in 6660 grid cells across the YRD providing hourly $PM_{2.5}$ measurements, resulting in potentially
excellent roles in constraining simulated PM$_{2.5}$ (Bocquet et al., 2015). In this study, we applied observed PM$_{2.5}$ concentrations
to constrain and evaluate the model performance. It is worth noting that the constraining capability of those observations varies
depending on specific configurations (e.g., the nature of the utilized DA method, the assimilation frequency, and the
representative errors of observations) (Bocquet et al., 2015; Chai et al., 2017; Ma et al., 2019; Rutherford, 1972). As shown in
Figure 1a, to consider regional impacts outside the YRD, the ground-level observations in the fan-shaped quadrilateral were
used to constrain the model performance. This was mainly due to the fact that this fan-shaped geographical scope covered
almost all key regions that had potentially regional impacts on the YRD, involving the Beijing-Tianjin-Hebei region (BTH),
the Pearl River Delta region, the Sichuan-Chongqing region, and the Shaanxi-Gansu region (Zhang et al., 2019). On the other
hand, the ground monitoring sites within the fan-shaped quadrilateral were significantly denser than those outside, thus leading
to much more effective DA results in practice (Bocquet et al., 2015; Chai et al., 2017). Collectively, to assimilate the
observations in the fan-shaped quadrilateral might be a sensible way to balance the DA effectiveness and the computing
efficiency. A resultant evidence lies in the model performance evaluation in Sect. 3.1, which would prove that this DA
configuration can enable reliable PM$_{2.5}$ simulations.

## 156 2.4 Optimal interpolation

Optimal interpolation (OI) was chosen to assimilate hourly observations into the WRF-CMAQ model, aiming to generate the
accurate state of spatiotemporal PM$_{2.5}$ distributions. Compared to the solely model-dependent results, this constraining method
relies on observations and thus makes it possible to minimize model uncertainties in optimizing the spatiotemporal PM$_{2.5}$
changes resulting from emission controls (Chai et al., 2017; Jung et al., 2019). The analysed states from the OI method were
calculated based on the following interpolation equation:
$$\mathbf{X}^{a} = \mathbf{X}^{b} + \mathbf{B}\mathbf{H}^{T}(\mathbf{H}\mathbf{B}\mathbf{H}^{T} + \mathbf{O})^{-1}(\mathbf{Y} - \mathbf{H}\mathbf{X}^{b}) \tag{1}$$

where $\mathbf{X}^{a}$ and $\mathbf{X}^{b}$ denote the analysis (constrained) and background (simulated) values, respectively. $\mathbf{B}$ and $\mathbf{O}$ are background
and observation error-covariance matrices, respectively, for which we assumed no correlation in this study. $\mathbf{H}$ refers to a
linearized observational operator, and $\mathbf{Y}$ represents the observation vector. The OI method is described in detail in Adhikary
et al. (Adhikary et al., 2008).
Once available measurements were assimilated, the states of the simulated variables were adjusted from their background
values to corresponding analysis states using the scaling ratio $\mathbf{X}^{a}/\mathbf{X}^{b}$ obtained following equation (1). As the measurements
were conducted at the surface, this ratio at each grid cell was used to scale all aerosol components below the boundary layer
top. Such simplification compensated for the lack of information to constrain speciated aerosol components or their vertical
distributions. When ground-level PM$_{2.5}$ measurements were assimilated, hourly observations were put into equation (1) to
construct the new analysis fields. All-day state variables associated with aerosols in the model were adjusted from their
background (simulated) to their analysis (constrained) states using the scaling factors ($\mathbf{X}^a/\mathbf{X}^b$). The adjusted model state
variables were then used to initiate the model to predict the next background state ($\mathbf{X}^b$) in Equation (1). Therefore, the
background state ($\mathbf{X}^b$) served as a prior model prediction before it was combined with the newly available observation ($\mathbf{Y}$) to
generate a new analysis state ($\mathbf{X}^a$) using Equation (1).
Measurements within the background-error correlation length scale were used to shape analysis states ($\mathbf{X}^a$). The background
error covariance $\mathbf{COV_{ij}}$ between any two grid cells $\mathbf{i}$ and $\mathbf{j}$ was simulated as

$$\mathbf{COV_{ij}} = \boldsymbol{\varepsilon_i}\boldsymbol{\varepsilon_j}\mathbf{e}^{-\frac{\Delta_{ij}}{L}} \tag{2}$$

where $\boldsymbol{\varepsilon_i}$ and $\boldsymbol{\varepsilon_j}$ referred to the standard deviations of the background errors in two grid cells and $\boldsymbol{\Delta_{ij}}$ denoted the distance
between the two grids. As a result, $\mathbf{L}$ was the background-error correlation length scale, which can be the Hollingsworth-
Lönnberg method (Chai et al., 2017; Hollingsworth and Lönnberg, 1986; Kumar et al., 2012). Figure 2 shows the correlation
coefficient, i.e., $\mathbf{COV_{ij}}/\boldsymbol{\varepsilon_i}\boldsymbol{\varepsilon_j}$, as a function of the separation distance between two grid cells, which was averaged over 10 km
bins. The results identified that a correlation length scale of ~ 180 km could be treated as the threshold. It allowed the
correlation coefficients to fall within the range of $\mathbf{e^{-1}}$, defining the effective radius of each individual observation. Due to the
intensive monitoring sites in our study domain, this threshold was applied uniformly for the YRD. In this study, observations
beyond the background-error correlation length scale would have no effect on $\mathbf{X}^a$. Following Chai et al. (Chai et al., 2017),
the standard deviation of the background errors was assigned as 60% of the background values, while the observational errors
were assumed to be $\pm$ 20% of the measurement values.

## 2.4 Experiment design

Anthropogenic emission controls and meteorological perturbations are both critical factors that dominate interannual and daily
variations in ambient PM$_{2.5}$ (Zhang et al., 2019). Our major objective is to isolate the impacts of emission-oriented long-term
and emergency measures and further explore the attainable PM$_{2.5}$ mitigation potential. We designed three sets of experiments,
which focused on three time periods, January 2016, January 2019, and the G20 period (from August 26, 2016 to September 7,
2016), respectively (Table 1).
For all experiments, the anthropogenic emissions were kept consistent (i.e., MEIC), while the ECMWF reanalysis datasets
accounted for the hourly observational constraints on spatiotemporal meteorological evolutions. The ECMWF reanalysis
datasets accounted for the hourly observational constraints on spatiotemporal meteorological evolutions. Therein almost all
necessary meteorological factors (nine variables), involving temperature, U wind component, V wind component, pressure,
relative humidity, precipitation, short-wave radiation, cloud cover, and planetary boundary layer height (PBLH), were
assimilated (https://apps.ecmwf.int/datasets/data/interim-full-daily/levtype=sfc/, last access: 7 March 2020). These
configurations unified both chemical (i.e., emission inventories) and meteorological input data for the WRF-CMAQ model.
Hence, the extent to which we introduce observational constraints on simulated $PM_{2.5}$ variations using the OI method is the
key to isolate the impacts of anthropogenic emission controls. Specifically, the differences in the constrained $PM_{2.5}$
concentrations between DA_2016 and DA_2019 reflected the net effects of anthropogenic emission controls and
meteorological perturbations between 2016 and 2019, while meteorological impacts therein were calculated as the
discrepancies in simulated $PM_{2.5}$ concentrations between NO_2016 and NO_2019 (Chen et al., 2019). Hence, by subtracting
meteorological impacts from the net effects, we can isolate the effects of anthropogenic emission controls attributable to the
long-term strategies.
The G20 summit provided a unique opportunity to realize the $PM_{2.5}$ mitigation potential in specific regions (Li et al., 2019a,
2017c; Ma et al., 2019; Shu et al., 2019; Yang et al., 2019). This is due to the fact that the Chinese government implemented
the most historically stringent, even unsustainable, strategies to curb anthropogenic emissions during that period in Hangzhou
and surrounding areas. To quantify the projected $PM_{2.5}$ abatement, we adopted the abovementioned method to constrain the
unique $PM_{2.5}$ variations in the DA_G20 experiment and further compared the corresponding results with those of the sole
model-dependent analysis (i.e., NO_G20). However, the subsequent discrepancies were related not only to the effects of
emergency anthropogenic emission strategies but also to the inherent biases mainly due to the emission inventory (Zhang et
al., 2019). In theory, such biases would generally remain unchanged in the short term when no emergency emission controls
occurred. Their consequent impacts could thus be stable under similar meteorological conditions. Therefore, to avoid additional
uncertainties, the adjacent periods of the G20 summit (i.e., pre- and post- periods, from August 11 to August 23, 2016 and
from September 18 to September 30, 2016, respectively) are the optimal alternative to eliminate the impacts of those inherent
biases. Figure S1 demonstrates the significantly similar meteorological fields among these three periods. As a result, the
corresponding experiments (i.e., DA_CON_G20 and NO_CON_G20) (Table 1) were conducted. By subtracting such
differences, we could isolate the $PM_{2.5}$ responses to the solely emergency anthropogenic emission strategies and finally achieve
the $PM_{2.5}$ mitigation potential for specific locations. Such localized $PM_{2.5}$ mitigation potential should be further expanded to
the entire YRD based on the impacts of both long-term and emergency strategies.
There is an essential prerequisite to above analysis. As the evaluation protocols, we need to verify that the DA experiments
(i.e., DA_2016, DA_2019, DA_G20, and DA_CON_G20) can reproduce the spatiotemporal variations in the $PM_{2.5}$ and major
meteorological fields (i.e., temperature, relative humidity, wind speed and air pressure) (Chai et al., 2017). While 244
monitoring stations reside in 6660 grid cells, 16 grid cells have two to three monitors in them. For these grid cells, only one
averaged measurement was used for DA. However, all the observations were compared against the constrained results.
Although SIA and SOA are key components of the ambient $PM_{2.5}$, extensive measurements at the regional scale of these
components are generally lacking. It is thus difficult to generate appropriate constraints on SIA and SOA (Chai et al., 2017;
Gao et al., 2017). Note that different anthropogenic emissions might lead to inconsistent estimation of meteorological effects
on ambient PM$_{2.5}$ (Chen et al., 2019). To eliminate this doubt, we conducted sensitivity tests by reducing MEIC with three
reasonable ratios (i.e., -5%, -25%, and -40%) over the YRD based on NO_2016 and NO_2019.

## 3 RESULTS

### 3.1 Data assimilation performance

Figure 3 shows spatial comparisons of hourly averaged concentrations of constrained and simulated PM$_{2.5}$ (i.e., the ones from
the cases with and without DA, respectively) with ground-level observations across the YRD for January 2016, January 2019,
and the G20 summit. In the NO_2016, NO_2019, and NO_G20 experiments, the simulated PM$_{2.5}$ concentrations generally
overestimated observed values by 16 ~ 57 μg/m$^3$, especially those in Hangzhou and surrounding areas during the G20 summit
(> 21 μg/m$^3$). Such prevailing overestimates were mainly a result of the anthropogenic emission inventory (i.e., MEIC), as a
bottom-up product, which notably cannot capture interannual emission changes since the base year 2012, as well as the large
emission controls resulting from the emergency controls during the G20 summit. By comparison, the constrained results
significantly approach observations. Specifically, in the DA_2016, DA_2019, and DA_G20 cases, the biases of the assimilated
PM$_{2.5}$ were all constrained in an extremely narrow range (i.e., 10 μg/m$^3$, 12 μg/m$^3$, and 13 μg/m$^3$, respectively), suggesting that
the DA method can reproduce the spatiotemporal distributions of surface PM$_{2.5}$ at the regional scale.
To achieve more targeted evaluations, it is necessary to further assess the ability of the DA method in reproducing the city-
level PM$_{2.5}$ responses. With the analysis of time series over the same periods, Figure 4 illustrates the comparisons between
hourly observed, simulated, and constrained PM$_{2.5}$ concentrations over the whole domain and four representative cities (i.e.,
Shanghai, Hangzhou, Nanjing, and Hefei). Similar to the spatial comparisons, the constrained PM$_{2.5}$ generally reproduces the
temporal variations in observations, while the model-dependent simulated results are prone to overestimating those
observations, in particular, the peaks by 85 ~ 257 μg/m$^3$.
As expected, basic evaluation indicators (i.e., the NMB and R values) of assimilated PM$_{2.5}$ exhibited significantly better
behaviour than those without constraints (Figure S2). Taking the simulated and assimilated results for Hangzhou during
January 2016 as an example, the corresponding R values improved from 0.63 to 0.98, while the NMB values were reduced
from 17% to 3%. Similar improvements, but with varying extent, were found in other paired experiments.
Owing to the fact that the distinct PM$_{2.5}$ levels might also play a potential role in the DA performance, we thus separated the
entire range of the observed PM$_{2.5}$ concentrations into four intervals (i.e., < 35 μg/m$^3$, 35 ~ 75 μg/m$^3$, 75 ~ 115 μg/m$^3$, and >
115 μg/m$^3$), exactly corresponding to the continuously increasing PM$_{2.5}$ levels. Figure S3 demonstrates that, relative to the sole
model-dependent configurations, this constraining method could substantially strengthen the model performance, especially

for the relatively elevated concentration intervals. Overall, the ranges of the NMB values and associated standard deviations decreased from -24 ~ 86% to -9 ~ 25% and 34 ~ 174 $\mu g/m^3$ to 12 ~ 52 $\mu g/m^3$, respectively. Theoretically, more frequent DA should lead to more robust simulations. Hourly observational constraints on the $PM_{2.5}$ concentrations were thus adopted to tackle this issue. This is the reason why the corresponding NMB values in the constraining cases roughly maintain stability, fluctuating over a narrow range (i.e., $\pm$ 20%) in the study periods (Figure S4). In addition, given that the assimilated ERA reanalysis dataset has much wider spatial coverage than ground-based measurements, we also reproduced the spatiotemporal variations in the meteorological factors (e.g., temperature, relative humidity, wind speed, and air pressure) (Figures S5 ~ S8). Together the comprehensive evaluation statistics as summarized in Tables S1 ~ S5, it has been demonstrated that the DA method can enable one to derive not only reliable $PM_{2.5}$ evolution but also accurate meteorological fields. Regional transport of $PM_{2.5}$ can thus be captured reasonably in this way.

### 3.2 Ambient $PM_{2.5}$ responses to the long-term strategies

The Chinese government has been implementing stringent emission control strategies since 2016, especially in the YRD (Feng and Liao, 2016; Li et al., 2019c). To quantify subsequent $PM_{2.5}$ responses is thus the prerequisite to our final objective, that is, to explore the associated $PM_{2.5}$ mitigation potential.

Interannual changes in spatiotemporal $PM_{2.5}$ distributions depended strongly on both anthropogenic emission controls and meteorological variations from 2016 to 2019. Their combined effects were reflected by the differences between the constrained results from DA_2016 and DA_2019. As shown in Figure 5a, such net impacts led to prevailing $PM_{2.5}$ abatement in the domain, especially in megacities, such as Shanghai (13 $\mu g/m^3$, 21%), Hangzhou (13 $\mu g/m^3$, 17%), Nanjing (6 $\mu g/m^3$, 8%), and Hefei (2 $\mu g/m^3$, 2%). In addition, noticeable $PM_{2.5}$ controls also occurred in the western and northern YRD, where abundant anthropogenic emissions are concentrated (Figure S9). Detailed differences are shown in Table S6.

Figure 5b highlights that the sole meteorological interferences played an extensively positive role in increasing the regional $PM_{2.5}$ concentrations for most areas of the domain (~ 12 $\mu g/m^3$, 15%). This also indirectly implied the importance of assimilating meteorology, which, however, were generally neglected by previous studies (Chen et al., 2019). In this study, we have eliminated this speculation. As shown in Figure S10 and Figure 5, even with the largest adjustment (i.e., -40%), such interferences could be well controlled within the 5% (< 3 $\mu g/m^3$) scope, let alone other tests (i.e., < 3%, < 2 $\mu g/m^3$). Moreover, these findings are consistent with previous analyses (Chen et al., 2019; Zhang et al., 2019). They generally reveal that reasonable changes in the bottom-up emissions, together with the same meteorology input data, would not remarkably alter the simulated results associated with meteorological effects on surface $PM_{2.5}$ (< 5%). As a result, some past studies even directly ignored such sensitivity tests without any discussion (Chen et al., 2019). Therefore, by subtracting those meteorological influences from the combined outcomes, we can finally derive the contributions of anthropogenic emission

controls to the PM$_{2.5}$ mitigation at the regional scale. Figure 5c illustrates that long-term emission control strategies from 2016 to 2019 produced substantial (> 14 μg/m$^3$, 19%) decreases in regional PM$_{2.5}$ concentrations, which are similar to those combined effects in terms of the spatial distributions.

For the entire domain, as well as the four representative cities, the synergy between anthropogenic emission controls and meteorological interferences on the PM$_{2.5}$ concentrations were calculated at the city level (Figure 6). We found that their net effects resulted in uniformly positive mitigations as follows: -2 μg/m$^3$ (-3%), -13 μg/m$^3$ (-21%), -12 μg/m$^3$ (-17%), -6 μg/m$^3$ (-8%), and -2 μg/m$^3$ (-3%) for the whole domain, Shanghai, Hangzhou, Nanjing, and Hefei, respectively, while the meteorological conditions therein offset such effects to different extents (5 ~ 18 μg/m$^3$, 16 ~ 24%). We recognized that the impacts of anthropogenic drivers on PM$_{2.5}$ concentrations in the southern and eastern parts of Zhejiang were evidently weaker than those in other regions in the YRD. This divergence can mostly be explained by spatial distributions of anthropogenic emissions. That is, anthropogenic emissions in the southern and eastern of Zhejiang were also significantly less than those in other regions (Figure S9), thus leading to substantially low PM$_{2.5}$ concentrations (Figure 3). Besides, meteorological fields in coastal regions, more conducive to PM$_{2.5}$ diffusion (Figure 5), might be another cause. The above findings confirmed that the PM$_{2.5}$ mitigation was dominated by anthropogenic emission controls, rather than meteorological variations. Furthermore, the corresponding spatiotemporal patterns were highly correlated to those of the anthropogenic emissions (Figure S9). This indicates that the impacts of the long-term strategies are mainly driven by anthropogenic emission mitigation.

### 3.3 Ambient PM$_{2.5}$ mitigation potential

The G20 summit offered a unique and ideal opportunity to clarify the effects of the most stringent emission control measures across the YRD from 2016 to 2019, which could be regarded as the localized PM$_{2.5}$ mitigation potential. Figure 7a shows the spatial differences between the constrained and simulated PM$_{2.5}$ concentrations, which were extracted from DA_G20 and NO_G20, for the period of the G20 summit. Inherent biases remained, primarily attributable to the priori anthropogenic emissions. Their subsequent impacts were then quantified by comparing the discrepancies between the results from two additional experiments (i.e., DA_CON_G20 and NO_CON_G20) (Figure 7b). More, such impacts were associated with relatively low standard deviations (< 5%), thus presenting a stably spatiotemporal state (Figure S11). This means that such estimations were also suitable for the G20 summit. Therefore, by subtracting them, the re-corrected differences would reflect the actual effects of the most stringent emission control measures for the G20 summit (Figure 7c). Such hotspots with extremely negative values reveal the dramatic PM$_{2.5}$ mitigations for these specific locations. The corresponding largest decreases in PM$_{2.5}$ concentrations (35 μg/m$^3$, 59%) occurred in Hangzhou and its surrounding areas, as expected. Following Hangzhou, other hotspots with relatively prominent declines also emerged in megacities, especially in Shanghai (32 μg/m$^3$, 51%), Nanjing (27 μg/m$^3$, 55%) and Hefei (24 μg/m$^3$, 44%). This behaviour could be explained by two inferences that: (i) local emission controls

in Hangzhou were projected to be conducted with the maximum execution efficiency compared to those in surrounding regions;
(ii) most of the emergency measurements generally targeted the vehicle and industry emissions that are clustered around the
urban rather than rural areas.
Compared to the long-term policies from 2016 to 2019, the emergency emission control measures implemented during the
G20 Summit achieved more significant decreases in $PM_{2.5}$ concentrations (17 $\mu g/m^3$ and 41%) over most of the whole domain,
especially in Hangzhou (24 $\mu g/m^3$, 48%) and Shanghai (21 $\mu g/m^3$, 45%) (Figure 8). Detailed differences are summarized in
Table S6.
To gain the regional $PM_{2.5}$ mitigation potential, (i) we first pinpointed the main urban areas of Hangzhou that covered 25 grid
cells (Figure S12), in which the most substantial $PM_{2.5}$ abatement, i.e., the localized $PM_{2.5}$ mitigation potential (> 22 $\mu g/m^3$
and > 59%) were identified. (ii) As the above hypothesis, the spatial distributions of the regional $PM_{2.5}$ mitigation potential
across the YRD were then assumed to follow those of the long-term strategy effects. (iii) Thus, by extrapolation in equal
proportion following such patterns and the localized $PM_{2.5}$ mitigation potential, we established the map of the $PM_{2.5}$ mitigation
potential across the YRD (Figure 9a). It should be noted that, as long as three premises, including typical weather backgrounds,
stable structures of anthropogenic emissions, and analogous emission control measures, remain unchanged, Figure 9a is a
reliably quantitative reference to characterize the attainable $PM_{2.5}$ abatement for the YRD in future.

**4 DISCUSSION**

The actual effectiveness of anthropogenic emission control measures, especially those directed at $PM_{2.5}$ mitigation, has long
been excluded from evaluation of air pollution policies in China, in part due to the complex synergy between anthropogenic
emissions and meteorology. Here, we provide a novel approach to explore the $PM_{2.5}$ responses to anthropogenic emission
control measures and their mitigation potential from 2016 to 2019 across the YRD, China. With the data assimilation method,
these estimates are projected to be highly reliable due to the sufficient observational constraints. The results demonstrate that
long-term anthropogenic emission control strategies from 2016 to 2019 have led to extensive impacts on $PM_{2.5}$ abatement
across the YRD, especially in the megacities, Shanghai, Hangzhou, Nanjing, and Hefei. In the context of the G20 summit, the
emergency strategies could achieve significant $PM_{2.5}$ abatement (> 50%) at specific locations, (i.e., urban Hangzhou),
representing the localized mitigation potential. By extrapolation based on the above results, we have established the first map
of the $PM_{2.5}$ mitigation potential for the YRD.
Numerous analyses have focused on Hangzhou during the G20 summit to detect impacts of emergency emission controls (Li
et al., 2019b, 2017c; Yu et al., 2018). However, previous analyses generally found more effective predictions (> 50%) at the
city level. This discrepancy might be related to the fact that such results were generally based on sole model-dependent
predictions, which are normally driven by uncertain bottom-up estimates of anthropogenic emissions. In addition, this study

addresses the YRD after 2016. Besides, similar opportunities also occurred at other spatiotemporal scales, such as the "APEC Blue" in 2014 and "Parade Blue" in 2015 over the BTH (Liu et al., 2016; Sun et al., 2016; Zhang et al., 2016). More aggressive achievements (> 55%) were generally attributed to emergency anthropogenic emission control measures (Sun et al., 2016). This might be related to the fact that, compared to the YRD, the BTH is associated with more abundant primary emissions (Zhang et al., 2019). The impacts of natural sources (e.g., biogenic emissions, wild fires, and natural dust) are not considered in this study. This is mainly because of two reasons. First, it has been widely demonstrated that biogenic emission changes are dominated by meteorological variations over a period of a few years (Wang et al., 2019). Moreover, the former is generally of minor significance for interannual $PM_{2.5}$ variations for the YRD (Mu and Liao, 2014; Tai et al., 2012). Second, satellite products, including MOD14 and AIRIBQAP_NRT.005 (https://worldview.earthdata.nasa.gov/), show that there was no noticeable wild fires and natural dust storms during this study period, thus allowing us to ignore the corresponding interferes. This study takes the advantage of observational constraints to gain the regional $PM_{2.5}$ mitigation potential. It could be further optimized by more extensive observations. Besides, extending the $PM_{2.5}$ mitigation potential in urban Hangzhou during the study period to the entire YRD in other time periods may introduce some uncertainties due to varying meteorology. As abovementioned, impacts of the extreme emergency emission controls are spatially inconsistent across the YRD. To explore regional $PM_{2.5}$ mitigation potential, it is thus unavoidable to extrapolate from local to regional scale. The consequent uncertainty mainly relates to the hypothesis that the spatial patterns of the $PM_{2.5}$ mitigation potential across the YRD should follow those of the impacts of the long-term emission control strategies. In addition, there are distinct DA methods (Bocquet et al., 2015). It is thus believed that replacing the OI with another DA algorithm would lead to slightly different results. Note that, as previous studies have demonstrated (Cheng et al., 2019; Zhai et al., 2019; Zhong et al., 2018), model uncertainties remain, although we have verified the constrained results. We have supplemented the additional discussions in Sect. 4 for further explanation. For instance, model simulations of aerosol components (e.g., sulfate and nitrate) are still poorly constrained. Moreover, they have not been evaluated due to lack of available observations. Yet previous studies find that the model tends to underestimate sulfate production during high RH (as pointed by the reviewer) as well as SOA (Li et al., 2017a; Wang et al., 2014; Zhong et al., 2018). As a result, these uncertainties can be propagated into the estimations of meteorological effects. Besides, like other atmospheric chemical transport models, the WRF-CMAQ model cannot provide model uncertainty information, while Monte Carlo simulations for complex CTMs would be unrealistic due to extremely high computation loadings (Zhong et al., 2018). Looking forward, continued advances in observational techniques, better understanding of chemical and meteorological processes, and their improved representations in CTMs are all factors that are critical to optimizing the estimates of the $PM_{2.5}$ mitigation potential.

## ASSOCIATED CONTENT

**Supporting Information.**

The supplement related to this article is available online.

## NOTES

The authors declare no competing financial interest.

## ACKNOWLEDGEMENTS

This study was supported by the Department of Science and Technology of China (No. 2016YFC0202702, 2018YFC0213506 and 2018YFC0213503), National Research Program for Key Issues in Air Pollution Control in China (No. DQGG0107) and National Natural Science Foundation of China (No. 21577126 and 41561144004). Pengfei Li is supported by Initiation Fund for Introducing Talents of Hebei Agricultural University (412201904) and Hebei Youth Top Fund (BJ2020032).

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

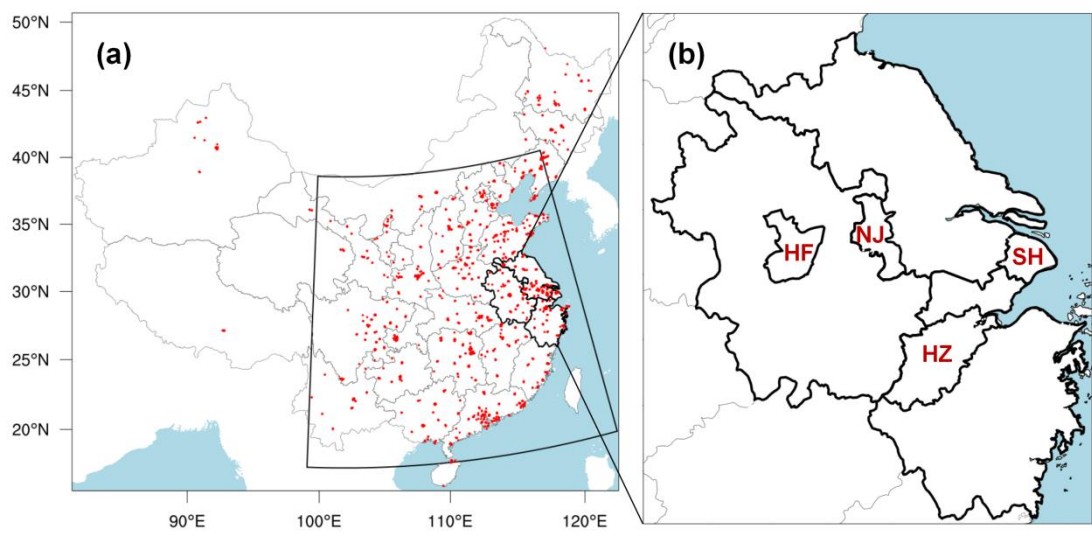

**Figure 1. (a) The model domain. Red dots denote the ground-level PM$_{2.5}$ measurements, which, within the**
**fan-shaped quadrilateral, are used to constrain the model predictions. (b) Black lines outline the boundaries**
**of the Yangtze River Delta (YRD), as well as four major cities considered (i.e., SH: Shanghai; HZ: Hangzhou;**
**NJ: Nanjing; HF: Hefei).**

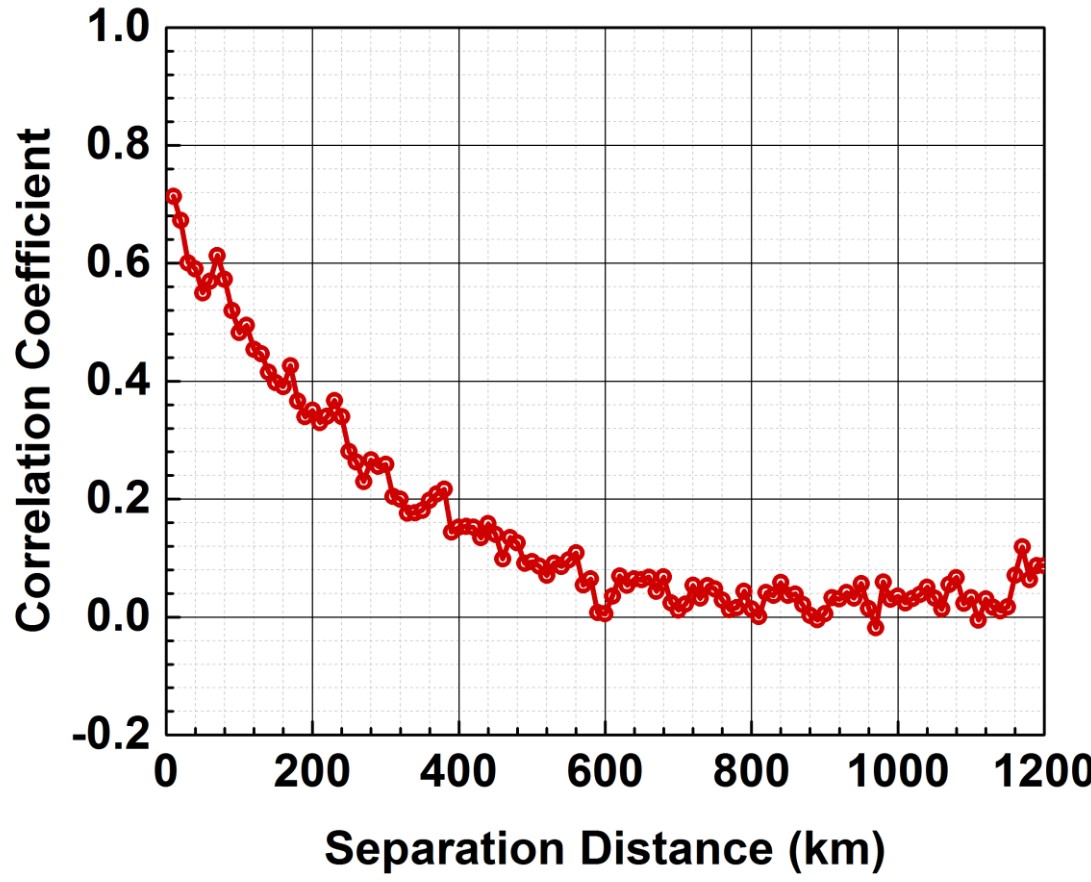


Figure 2. Correlation coefficients (averaged over 10 km) as a function of the separation distances between two

surface-level monitoring stations using the Hollingsworth-Lönnberg method.

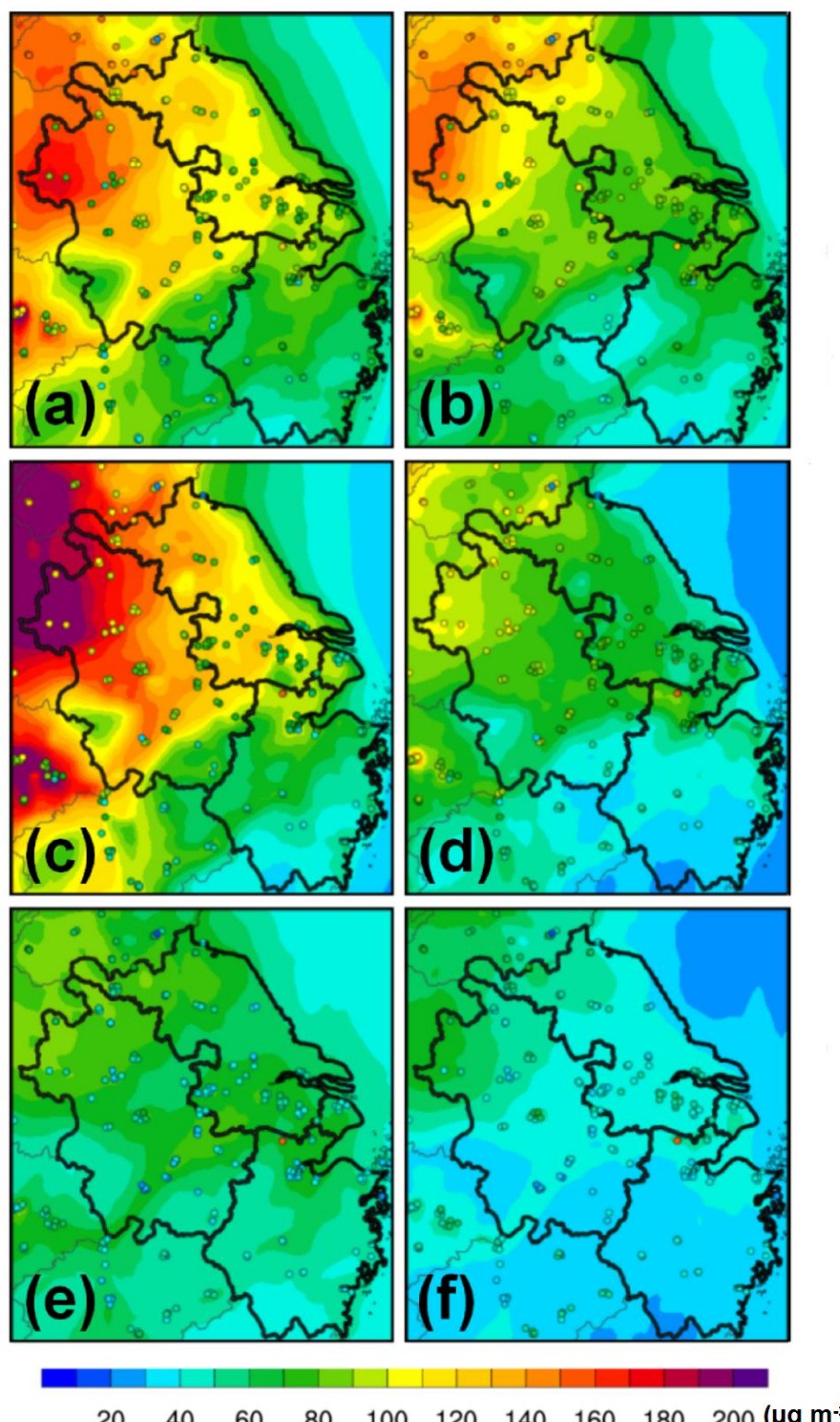

**Figure 3. Spatial comparisons of hourly-averaged concentrations of simulated and constrained PM$_{2.5}$ with**
**surface observations across the YRD for January 2016 (top panel), January 2019 (middle panel), and the G20**
**summit (bottom panel): (a) NO_2016; (b) DA_2016; (c) NO_2019; (d) DA_2019; (e) NO_G20; (f) DA_G20.**
**Circles denote ground measurement sites.**

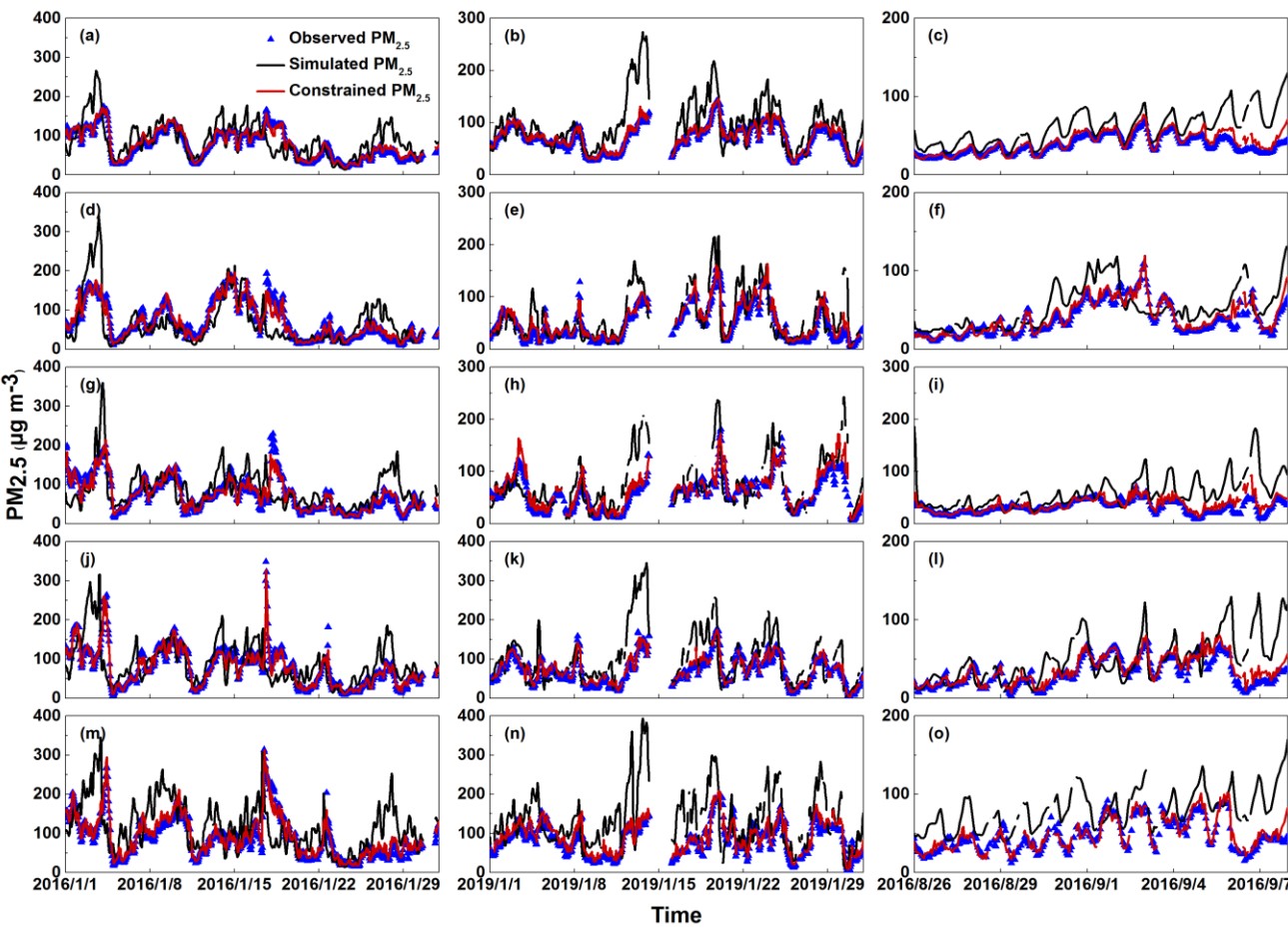


**Figure 4.** Time series of the comparisons between hourly observed, simulated, and constrained PM2.5 concentrations for January 2016 (left column), January 2019 (middle column), and the G20 summit (right column) over (a – c) the whole domain as well as in four representative cities, which are as follows: (d - f) Shanghai, (g - i) Hangzhou, (j - l) Nanjing, and (m - o) Hefei. The black circles, black lines, and red lines denote the hourly observed, simulated, and constrained PM2.5 concentrations, respectively.

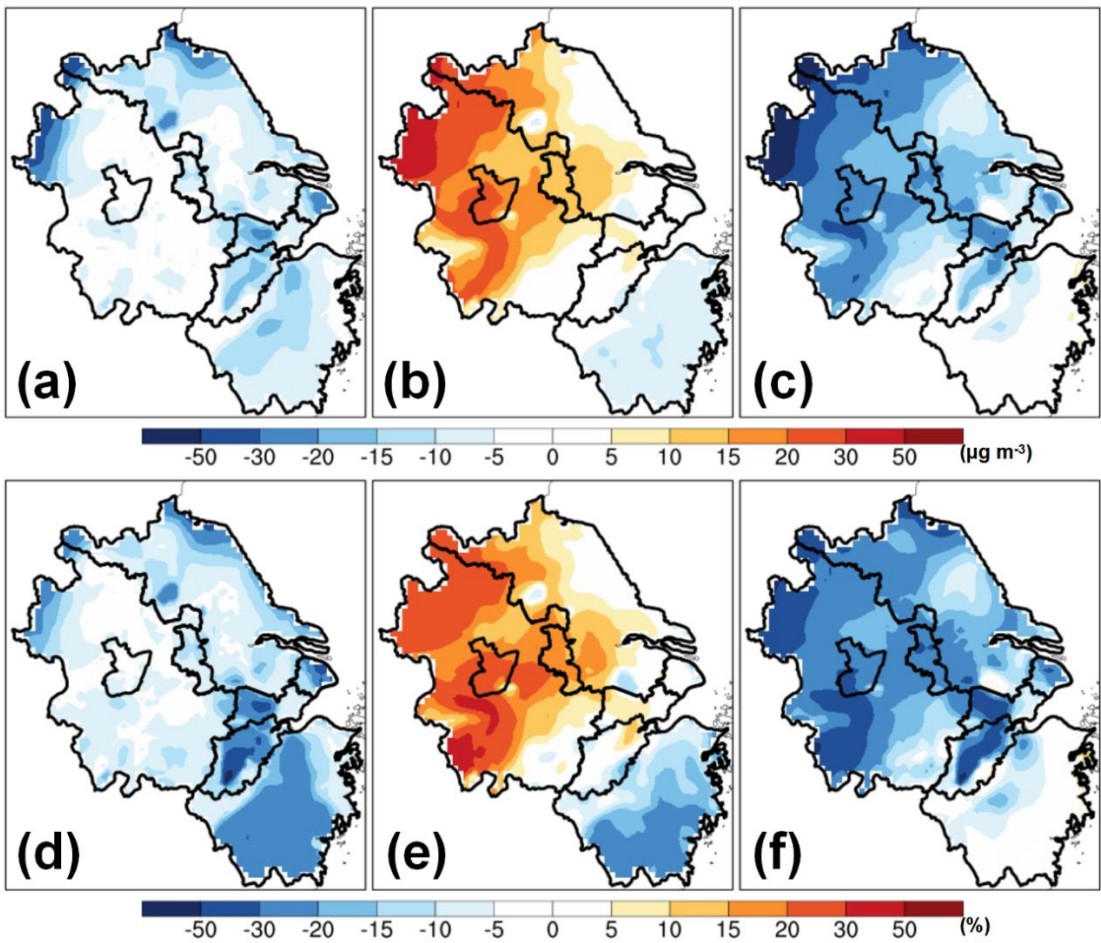

**Figure 5. The impacts of anthropogenic emission controls and meteorological variations on spatial PM$_{2.5}$ concentrations in January from 2016 to 2019. (a, d) Their net impacts. (b, e)**
**meteorological impacts. (c, f) the impacts of anthropogenic emission controls. The top and bottom panels refer to the changes in absolute values and relative percentages, respectively.**

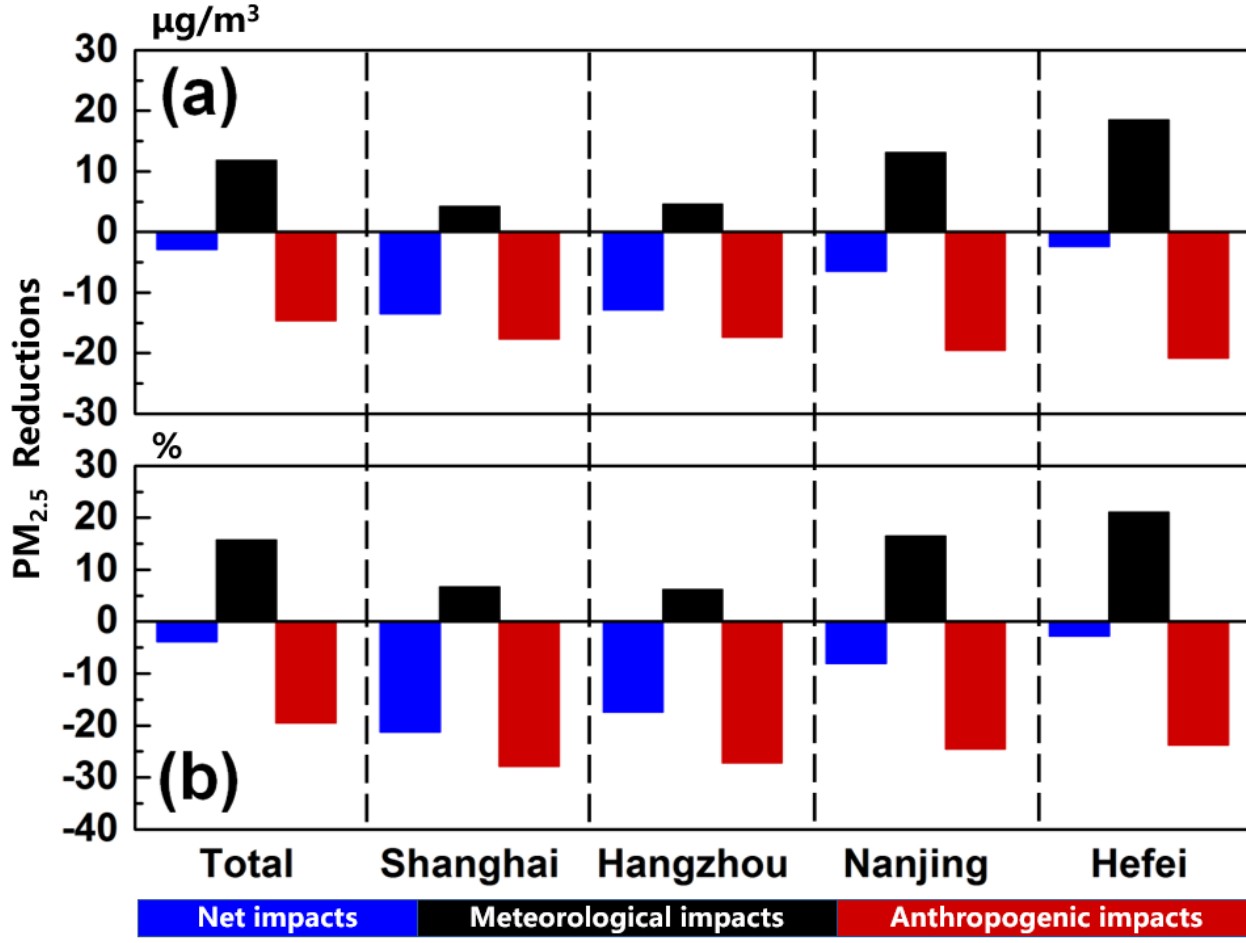


**Figure 6. The impacts of anthropogenic emission controls and meteorological variations on PM₂.₅ concentrations in January from 2016 to 2019 over the whole domain as well as in four**


**representative cities (i.e., Shanghai, Hangzhou, Nanjing, and Hefei). The top and bottom panels refer to the changes in absolute values and relative percentages, respectively.**


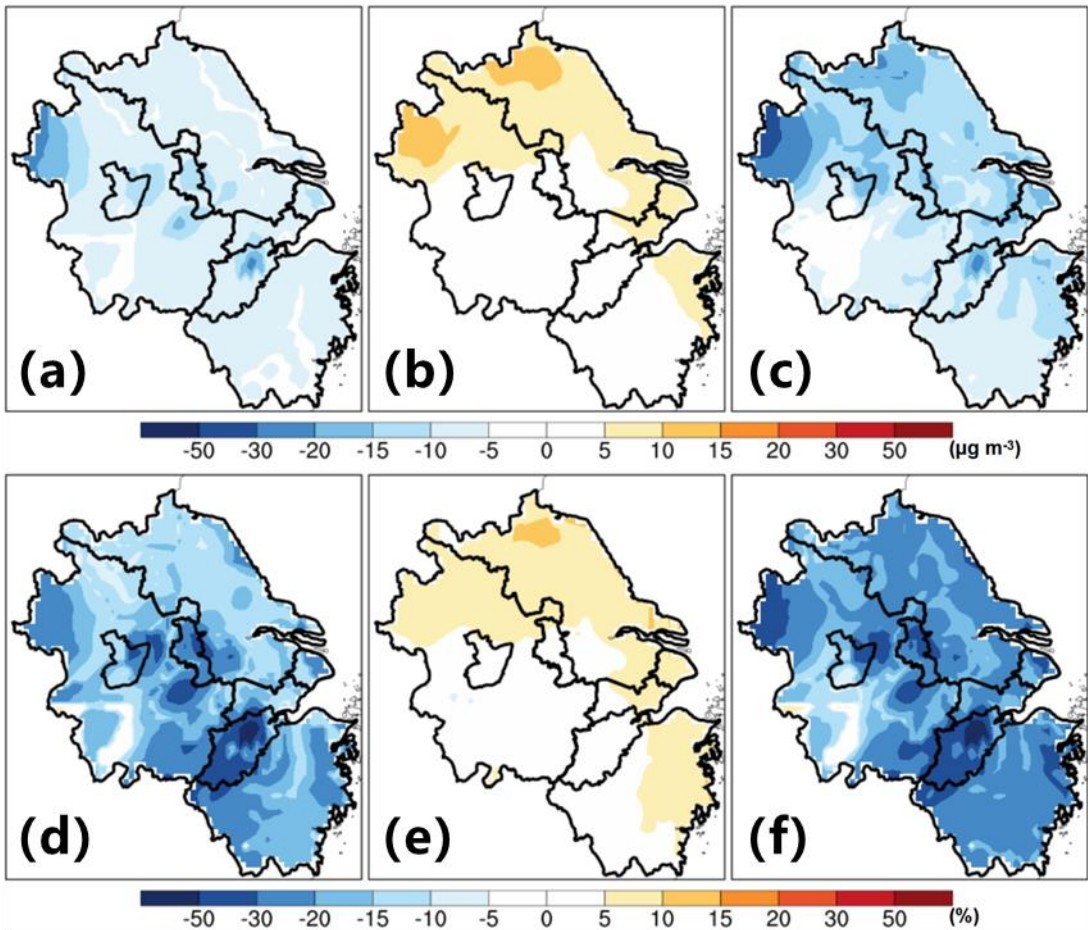


**Figure 7. The impacts of anthropogenic emission controls and inherent biases on spatial PM$_{2.5}$ concentrations during the G20 summit. (a, d) Their net impacts. (b, e) the impacts of inherent biases. (c, f) the impacts of anthropogenic emission controls. The top and bottom panels refer to the changes in absolute values and relative percentages, respectively. Inherent biases are mainly due to the prior anthropogenic emissions.**

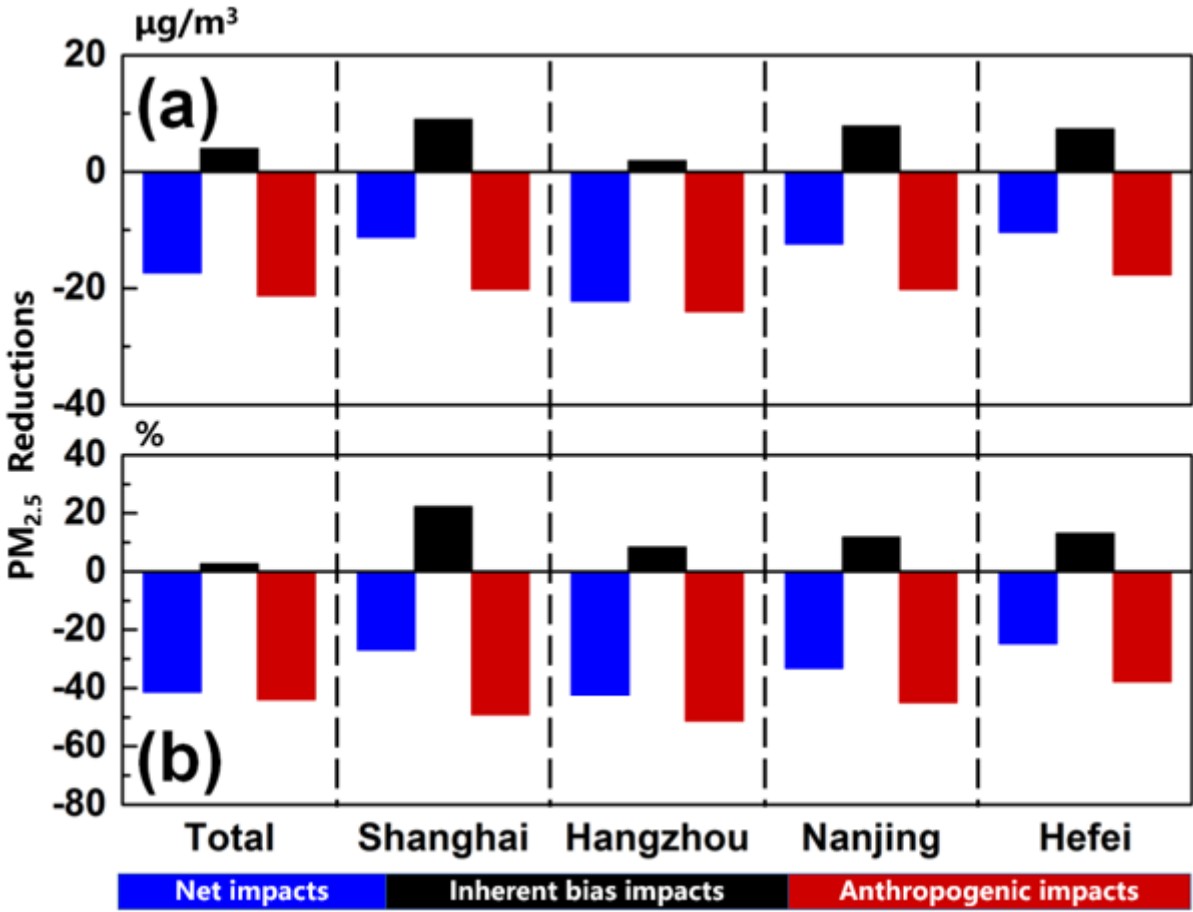


**Figure 8. The impacts of anthropogenic emission controls and inherent biases on PM2.5 concentrations during the G20 summit over the whole domain as well as in four representative**

**cities (i.e., Shanghai, Hangzhou, Nanjing, and Hefei). The top and bottom panels refer to the changes in absolute values and relative percentages, respectively. Inherent biases are**

**mainly due to the prior anthropogenic emissions.**

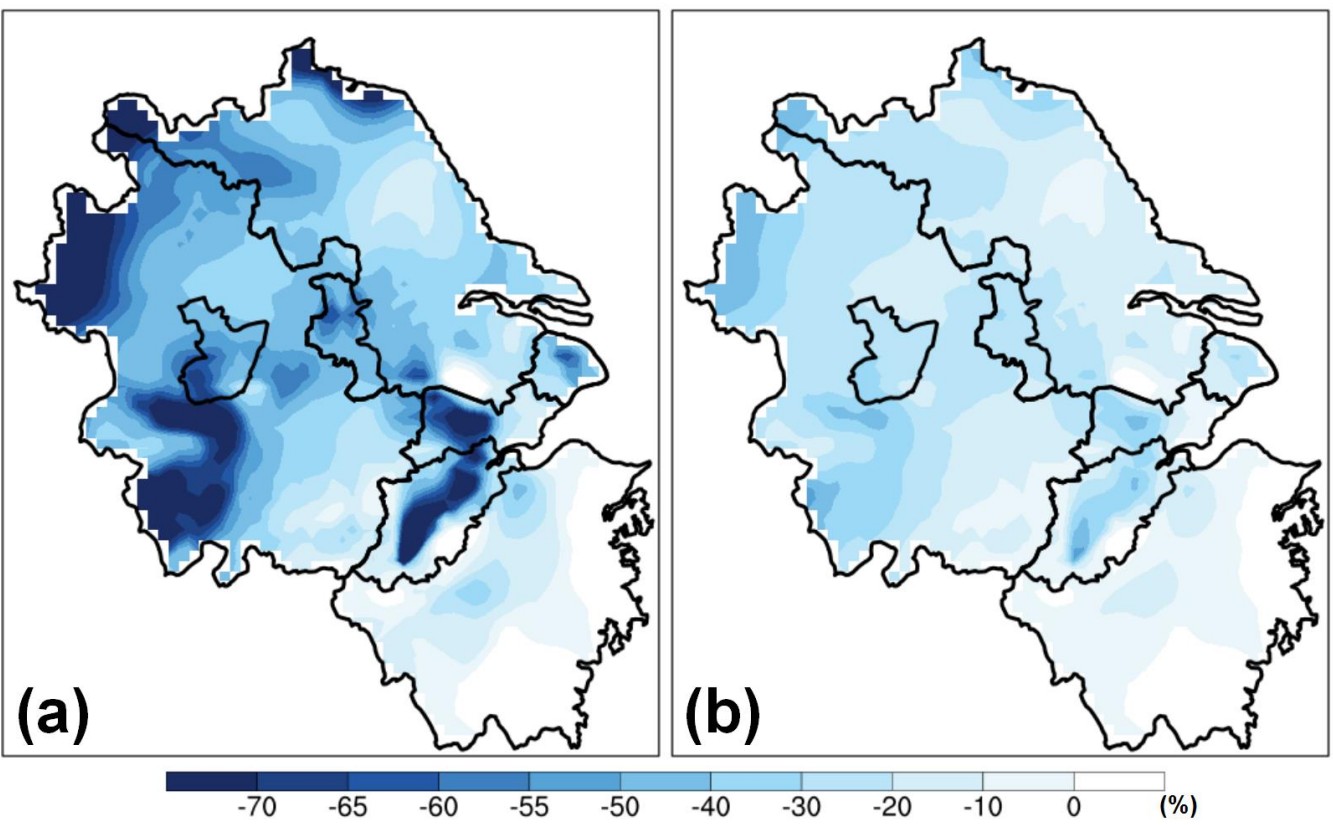


Figure 9. (a) Spatial distributions of the PM$_{2.5}$ mitigation potential across the YRD and (b) their differences with the impacts of long-term emission control strategies from 2016 to 2019
(Fig. 5f). Both spatial patterns of long-term emission control strategy impacts (Fig. 5f) and the localized PM$_{2.5}$ mitigation potential in the main urban areas of Hangzhou (Fig. S10), with
the proportion calculator, result in Fig. 9a.



**Table 1. The experiments to isolate the effects of anthropogenic emission controls due to the long-term and emergency emission control strategies.**

| Experiments | Time Periods | Priori Anthropogenic Emissions | Constrained Meteorology | Constrained Observations | Comparisons and Purposes |
|---|---|---|---|---|---|
| DA_2016 | January 2016 | MEICv1.2 | Yes | Yes | The net effects of major driving factors (i.e., anthropogenic emission controls and meteorological variations) from 2016 to 2019. |
| DA_2019 | January 2019 | | Yes | Yes | |
| NO_2016 | January 2016 | MEICv1.2 | Yes | No | The effects of meteorological variations from 2016 to 2019. |
| NO_2019 | January 2019 | | Yes | No | |
| DA_G20 | from August 26 to September 7, 2016 | MEICv1.2 | Yes | Yes | The net effects of major driving factors (i.e., anthropogenic emission controls and the uncertainties in the priori anthropogenic emissions) during the G20 summit. |
| NO_G20 | | | Yes | No | |
| DA_CON_G20 | from August 11 to August 23 and from September 18 to September 30, 2016 | MEICv1.2 | Yes | Yes | The effects of the uncertainties in the priori anthropogenic emissions. |
| NO_CON_G20 | | | Yes | No | |
