# Peer review of "Significant wintertime PM2.5 mitigation in the Yangtze River Delta,"

_Atmospheric Chemistry and Physics, 2020_

## Referee Comment (RC1) · Anonymous Referee #2 · 19 Aug 2020

This paper uses a data assimilation method to constrain the modeled PM2.5 concentrations over the Yangtze River Delta (YRD) region, and distinguish the impact on PM2.5 from meteorology and emission variations. The results show that the emission reduction measures in G20 summit and long-term emission control strategies in YRD successfully curb the PM2.5 levels both locally and regionally. This paper is good in general and within the scope of Atmospheric Chemistry and Physics. I recommend for publication once the specific comments expressed below are addressed. Specific comments: 1) The author should provide more details regarding how to conduct data

assimilation. First, the author needs to perform a sensitivity analysis in order to proof that choosing the fan-shaped quadrilateral (Figure1ïijĽminimizes the impact from outside on the YRD region. Second, how is the modeled PM2.5 constrained spatiotemporally by observations, applying DA generated scaling factors to the whole fan-shaped quadrilateral region, the YRD region, city by city, or grid by grid, and hour by hour or day by day? 2) The author used a statistical method to establish the correlation coefficients and chose separation distance of 180km as a threshold. The author needs to give more explanations on the value of chosen. If the purpose is to find a correlation length scale to minimize the effect on Xa, based on Fig 2, it seems that separation distance of 600km would be more appropriate. 3) How did the author isolate the impact from emission reductions on PM2.5 concentrations? Did the author use the constrained PM2.5 subtract the impact on simulated PM2.5 from meteorological variations? Even the modeled temperature, humidity, wind speed, and air pressure were also assimilated in this study, there are other parameters, for example, modeled PBL height, causing large uncertainties in the modeled meteorological field, and thus leading to bias and error in the calculated net impacts from emission variations. For example, figures c and f in Fig 5, show very small impact of anthropogenic emission control from 2016 to 2019 in most of Zhejiang province compared to the other provinces in the YRD region. Is it reasonable? 4) How did the author consider the regional transport of PM2.5 in this study? The regional emission control effect on PM2.5 may have influence on calculated net impact of emission reduction in each city and the localized mitigation potential.

---

## Referee Comment (RC2) · Anonymous Referee #1 · 20 Aug 2020

Want et al. evaluated the effect of long-term and emergent emission control strategies on the PM2.5 levels in Yangtze River Delta of China, by combining modeling analysis with observations. They found the decline in PM2.5 concentration during 2016-2019 was mainly due to emission control. The decline would be even greater if the meteorology was not unfavorable. Great potential of further decrease is manifested in analysis of data during G20 period when short-term emergent measures were taken. The discussion is valuable for assessment of past policies and design of future ones. However, to be publishable in ACP, the current manuscript requires further improvement.

[Figure]

Particularly, inadequate credits are given to the existing literature that performed similar analysis of separating meteorology and emission effects on recent PM2.5 trend in China. Instead, the authors tried to impress the reader by suggesting that this study is the first to do so. Just to list a few studies in literature (and I believe there might be more), Zhang et al., 2019. Drivers of improved PM2.5 air quality in China from 2013 to 2017, PNAS Zhai et al., 2019. Fine particulate matter (PM2.5) trends in China, 2013–2018: separating contributions from anthropogenic emissions and meteorology, ACP Zhong et al., 2018. Distinguishing Emission-Associated Ambient Air PM2.5 Concentrations and Meteorological Factor-Induced Fluctuations, EST The authors should review the existing literature and put emphasize on their innovations.

I also have concerns about the methodology. Assimilation is used for calculating the total effect (emis+met) which gives a good representation of PM2.5 distributions, despite any model errors. But assimilation cannot be used for calculating met-only effect. Therefore, model errors may propagate into the met effect. I wonder what uncertainties this inconsistency in two pairs of simulations would cause for the results. The authors evaluated model emissions and concluded the impact is small. But it is not shown if other model errors may be significant. For example, studies have found that model tends to underestimate sulfate production during high RH in China. More evaluation of the model performance may be useful for interpreting the result.

The inclusion of short-term G20 period is interesting. But I am not completely convinced that the mitigation potential map is useful at all. At a first glance, the map does not seem to be very different from conducting a zero-ish YRD emission simulation with the model and then do a subtraction. The problem is that the authors did not provide information about (1) what types or fractions of emissions were shut down during the event; (2) is the emission shutdown implemented in Hangzhou, or Zhejiang, or YRD? Without this information, it is not possible to interpret the mitigation potential.

In addition, the writing needs to be improved throughout the text. Some word choices aren't proper, and in the comments below I picked out some. Some descriptions of the

methods need to be clarified as well.

Specific comments Line 41: Not clear from the text whether "> 14 $\mu$g/m3, 19%" is PM2.5 levels, or in fact, reduction in PM2.5 concentrations. Please clarify.

Line 42-44: Confusing, as it interrupts the flow and misleads a reader that the decline in Hangzhou (35 ug/m3) is due to G20 control measures. I suggest moving the sentence to either Line 40 after "YRD, China" or to Line 48 before "Compared to the long-term policies..."

Line 46: remove "thus"

Line 99: should -> can

Line 105: "unprecedented" is a too-big word here that I suggest to remove. Same for other occurrences of the word in the paper.

Line 125: Be consistent with the citation format.

Line 137-139: Are meteorological observations assimilated in addition to chemical observations. If so, describe the meteorological observations that are assimilated. If not, I don't think it is sufficient to just use initial and boundary conditions from reanalysis data. The WRF should be run in a nudging mode, so the meteorology is close to reality.

Line 155: "Prior anthropogenic emissions"? Do you optimize emissions at all? I cannot find such description throughout the text. If not, it should not be called "prior".

Line 166: "more" is not a proper conjunction word in formal English writing.

Line 176: grids -> grid cells

Line 176: "potential excellent roles" Rephrase it.

Section 2.4: What is the assimilation window? Daily? hourly? Line 212: "the threshold pinpointing the key value of the 213 correlation coefficients (e−1)". -> e-folding length

Line 217-219: well, it is still static in the relative sense. I don't this means anything. I'd suggest removing this statement.

Line 231: "unify the chemical inputs for the WRF-CMAQ model". What does this mean?

Line 237-240: Up to this point, we still do not know what method the authors use to separate effects of meteorology and emission. A clear description of the method is needed before this point.

Line 257: It is cursory to conclude these three periods have similar meteorology based on Fig. S1. The validity of the analysis is relied on the assumption that they are similar. E.g., one factor that is not analyzed is wind direction. Showing maps of circulation pattern will also help.

Fig. 4. Is Fig. 4 useful? It is no surprising that the assimilated simulation could better reproduce observations, which are used in assimilation. It means nothing.

Line 307-311. There is a jump in the logic of this sentence. I'd remove it.

Line 350-352: Many studies have properly separated the effects from meteorology and emissions, though with different approaches. I don't think the statement is fair.

Line 353: It's unclear to me whether 5% and 3% are relative to mean PM2.5 concentration or mean reduction of PM2.5. Be more explicit.

Line 355: how do you prove it was "under the same meteorological condition".

Line 370: what is "concurrent meteorology"? better to rephrase it.

Line 375-376: Of course the long-term strategies is emission oriented. You cannot change weather easily after all... I guess the author wanted to say the long-term decrease in PM2.5 was driven mainly by decreased emissions.

Line 387: what is "stably spatiotemporal state". Rephrase it.

Line 412: remove "unprecedented".

Line 414: what is "stable supply-side structures"? Not directly related to air quality to me.

Line 418: There have been quite a few papers discussing the effect of emission control strategy. To list a few, Zhang et al., 2019. Drivers of improved PM2.5 air quality in China from 2013 to 2017, PNAS Zhai et al., 2019. Fine particulate matter (PM2.5) trends in China, 2013–2018: separating contributions from anthropogenic emissions and meteorology, ACP Zhong et al., 2018. Distinguishing Emission-Associated Ambient Air PM2.5 Concentrations and Meteorological Factor-Induced Fluctuations, EST Although the method may not be the same, the authors should give credit to these studies rather than claim this is the first study trying to separate effects of met and emission.

Line 444: "rudimentary" may not be a proper word here.

Line 449: the statement that "the biogenic emissions are unimportant for IAV of PM2.5" may be true for YRD, but may not be "generally" true for elsewhere in the world. I'd suggest being more specific.
* * *

---

## Author Comment (AC1) · 8 Sep 2020

**General comments:**

This paper uses a data assimilation method to constrain the modelled PM$_{2.5}$ concentrations over the Yangtze River Delta (YRD) region and distinguish the impact on PM$_{2.5}$ from meteorology and emission variations. The results show that the emission reduction measures in G20 summit and long-term emission control strategies in YRD successfully curb the PM$_{2.5}$ levels both locally and regionally. This paper is good in general and within the scope of Atmospheric Chemistry and Physics. I recommend for publication once the specific comments expressed below are addressed.

**Response:** We thank the reviewer for the thoughtful comments on our paper and have addressed these specific comments as below.

**Specific comments:**

1. The author should provide more details regarding how to conduct data assimilation. First, the author needs to perform a sensitivity analysis in order to proof that choosing the fan-shaped quadrilateral (Figure 1a) minimizes the impact from outside on the YRD region. Second, how is the modelled PM$_{2.5}$ constrained spatiotemporally by observations, applying DA generated scaling factors to the whole fan-shaped quadrilateral region, the YRD region, city by city, or grid by grid, and hour by hour or day by day?

**Response:** Thanks. We have supplemented the additional discussions in Sect. 2.3 to explain why we choose the ground-level observations within the fan-shaped quadrilateral to constrain the model performance. As pointed by the reviewer, we aim to minimize the impacts outside the YRD region. Specifically, this was mainly due to the fact that this fan-shaped geographical scope covered almost all key regions that had potentially regional impacts on the YRD, involving the Beijing-Tianjin-Hebei region (BTH), the Pearl River Delta region, the Sichuan-Chongqing region, and the Shaanxi-Gansu region (Zhang et al., 2019). On the other hand, the ground monitoring sites within the fan-shaped quadrilateral were significantly denser than those outside, thus leading to much more effective DA in practice (Bocquet et al., 2015; Chai et al., 2017). Therefore, to assimilate the observations within the fan-shaped quadrilateral might be a sensible way to balance the DA effectiveness and computing efficiency. A resultant evidence lies in the model performance evaluation in Sect. 3.1, which would prove that this DA

configuration can enable reliable PM$_{2.5}$ simulations. Collectively, we might eliminate the need of the associated sensitivity analysis.

In addition, we have supplemented the more discussions in Sect. 2.4 to further detail how to conduct observational constraints on the model simulations spatiotemporally. In short, we conducted hourly DA for grid cells. Note that the effective radius of each individual observation should be calculated in advance. When ground-level PM$_{2.5}$ measurements were assimilated, hourly observations were put into equation (1) to construct the new analysis fields. All-day state variables associated with aerosols in the model were adjusted from their background (simulated) to their analysis (constrained) states using the scaling factors $(\mathbf{X}^a/\mathbf{X}^b)$. The adjusted model state variables were then used to initiate the model to predict the next background state $(\mathbf{X}^b)$ in Equation (1). Therefore, the background state $(\mathbf{X}^b)$ served as a prior model prediction before it was combined with the newly available observation $(\mathbf{Y})$ to generate a new analysis state $(\mathbf{X}^a)$ using Equation (1).

Measurements within the background-error correlation length scale were used to shape analysis states $(\mathbf{X}^a)$. The background error covariance $\mathbf{COV_{ij}}$ between any two grid cells $\mathbf{i}$ and $\mathbf{j}$ was simulated as

$$\mathbf{COV_{ij}} = \boldsymbol{\varepsilon_i}\boldsymbol{\varepsilon_j}\mathbf{e}^{-\frac{\boldsymbol{\Delta_{ij}}}{\mathbf{L}}} \tag{2}$$

[revised manuscript text omitted]

80    2. The author used a statistical method to establish the correlation coefficients and chose separation distance of 180 km as a threshold. The author needs to give more explanations on the value of chosen. If the purpose is to find a correlation length scale to minimize the effect on $\mathbf{X}^a$, based on Fig 2, it seems that separation distance of 600 km would be more appropriate.

**Response:** Thanks. The objective of identifying the background-error correlation length scale is to define the effective radius of each individual observation and thus to establish reliable analysis states ($\mathbf{X}^a$). Here the Hollingsworth-Lönnberg approach,

85    wildly used for decades (Chai et al., 2017; Hollingsworth and Lönnberg, 1986; Kumar et al., 2012), is applied to calculate the background-error correlation length scale. Observations beyond the background-error correlation length scale were assumed to have no effect on $\mathbf{X}^a$. Once observations far away are introduced, more background errors $\mathbf{COV_{ij}}$ , larger than $\mathbf{e}^{-1}$, would be put into $\mathbf{X}^a$ as calculated in Equation (2). Corresponding detailed information has been given in the response for the specific comment (2).

90

3. How did the author isolate the impact from emission reductions on $PM_{2.5}$ concentrations? Did the author use the constrained $PM_{2.5}$ subtract the impact on simulated $PM_{2.5}$ from meteorological variations? Even the modelled temperature, humidity, wind speed, and air pressure were also assimilated in this study, there are other parameters, for example, modelled PBL height, causing large uncertainties in the modelled meteorological field, and thus leading to bias and error in the calculated net impacts

95    from emission variations. For example, figures c and f in Fig 5, show very small impact of anthropogenic emission control from 2016 to 2019 in most of Zhejiang province compared to the other provinces in the YRD region. Is it reasonable?

**Response:** Thanks. Yes, it is reasonable. We isolated anthropogenic impacts on PM$_{2.5}$ concentrations by subtracting the corresponding meteorological impacts from the constrained PM$_{2.5}$ fields. To further illustrate the process of meteorological assimilations, we have supplemented the additional discussions in Sect. 2.4. The ECMWF reanalysis datasets accounted for the hourly observational constraints on spatiotemporal meteorological evolutions. Therein almost all necessary meteorological factors (nine variables), involving temperature, U wind component, V wind component, pressure, relative humidity, precipitation, short-wave radiation, cloud cover, and boundary layer height, were assimilated (https://apps.ecmwf.int/datasets/data/interim-full-daily/levtype=sfc/, last access: 7 March 2020).

The model evaluation provides a more direct way to verify the corresponding model performance. As highlighted in Sect. 3.1, given the fact that the assimilated ERA reanalysis dataset has much wider spatial coverage than ground-based measurements, we also reproduced the spatiotemporal variations in the meteorological factors (e.g., temperature, relative humidity, wind speed, and air pressure) (Figures S5 ~ S8). Together with the comprehensive evaluation statistics as summarized in Tables S1 ~ S5, it has been demonstrated that the DA method can enable one to derive not only reliable PM$_{2.5}$ evolution but also accurate meteorological fields.

In terms of the issue associated with Zhejiang, we have supplemented the additional interpretations in Sect. 3.2. The impacts of anthropogenic drivers on PM$_{2.5}$ concentrations in the southern and eastern parts of Zhejiang were evidently weaker than those in other regions in the YRD. This divergence can mostly be explained by spatial distributions of anthropogenic emissions. Anthropogenic emissions in the southern and eastern parts of Zhejiang were also significantly less than those in other regions (Figure S9), thus leading to substantially low PM$_{2.5}$ concentrations (Figure 3). Besides, meteorological fields in the coastal regions, more conducive to PM$_{2.5}$ diffusions (Figure 5), might be another cause.

**Added/rewritten part in Sect. 2.4:** For all experiments, the prior anthropogenic emissions were kept consistent (i.e., MEIC), while the ECMWF reanalysis datasets accounted for the hourly observational constraints on spatiotemporal meteorological evolutions. The ECMWF reanalysis datasets accounted for the hourly observational constraints on spatiotemporal meteorological evolutions. Therein almost all necessary meteorological factors (nine variables), involving temperature, U wind component, V wind component, pressure, relative humidity, precipitation, short-wave radiation, cloud cover, and planetary boundary layer height (PBLH), were assimilated (https://apps.ecmwf.int/datasets/data/interim-full-daily/levtype=sfc/, last access: 7 March 2020).

**Added/rewritten part in Sect. 3.1:** In addition, given the fact that the assimilated ERA reanalysis dataset has much wider spatial coverage than ground-based measurements, we also reproduced the spatiotemporal variations in the meteorological factors (e.g., temperature, relative humidity, wind speed, and air pressure) (Figures S5 ~ S8). With the comprehensive evaluation statistics as summarized in Tables S1 ~ S5, it has been demonstrated that the DA method can enable one to derive not only reliable PM$_{2.5}$ evolutions but also accurate meteorological fields.

**Added/rewritten part in Sect. 3.2:** We recognized that the impacts of anthropogenic drivers on PM$_{2.5}$ concentrations in the southern and eastern parts of Zhejiang were evidently weaker than those in other regions in the YRD. This divergence can mostly be explained by spatial distributions of anthropogenic emissions. Anthropogenic emissions in the southern and eastern

of Zhejiang were also significantly less than those in other regions (Figure S9), thus leading to substantially low PM$_{2.5}$ concentrations (Figure 3). Besides, meteorological fields in the coastal regions, more conducive to PM$_{2.5}$ diffusions (Figure 5), might be another cause.

135    4. How did the author consider the regional transport of PM$_{2.5}$ in this study? The regional emission control effect on PM$_{2.5}$ may have influence on calculated net impact of emission reduction in each city and the localized mitigation potential.

**Response:** Thanks. We agree with the reviewer that regional transport of PM$_{2.5}$ is central to our results and thus have considered it carefully. Using observational constraints on the state-of-the-art model, we have reproduced spatiotemporal variations in both PM$_{2.5}$ and meteorological factors, as illustrated in Sect. 3.1, and thus derived the reliable estimations of

140    regional transport of PM$_{2.5}$. Hence, we have supplemented a sentence in Sect. 3.1 to highlight this point.

Considering the main objective of this work, we have not conducted source apportionments to predict the impacts of regional transport of PM$_{2.5}$. In theory, regional transport of PM$_{2.5}$ can be attributable to both anthropogenic and meteorological drivers. In turn, we provide paired experiment designs to isolate anthropogenic impacts by subtracting meteorological perturbations (i.e., the differences in simulated PM$_{2.5}$ concentrations between NO_2016 and NO_2019 and between DA_CON_G20 and

145    NO_CON_G20) from the constrained PM$_{2.5}$ fields (i.e., DA_2016 and DA_2019 / DA_G20).

**Added/rewritten part in Sect. 3.1**: Regional transport of PM$_{2.5}$ can thus be captured reasonably in this way.

---

## Author Comment (AC2) · 8 Sep 2020

General comments: Wang et al. evaluated the effect of long-term and emergent emission control strategies on the PM2.5 levels in Yangtze River Delta of China, by combining modelling analysis with observations. They found the decline in PM2.5 concentration during 2016-2019 was mainly due to emission control. The decline would be even greater if the meteorology was not unfavorable. Great potential of further decrease is manifested in analysis of data during G20 period when short-term emergent measures were taken. The discussion is valuable for assessment of past policies and design of

future ones. However, to be publishable in ACP, the current manuscript requires further improvement.

Response: We thank the reviewer for the thoughtful comments on our paper and have addressed them as below.

1. Particularly, inadequate credits are given to the existing literature that performed similar analysis of separating meteorology and emission effects on recent PM2.5 trend in China. Instead, the authors tried to impress the reader by suggesting that this study is the first to do so. Just to list a few studies in literature (and I believe there might be more), Zhang et al., 2019. Drivers of improved PM2.5 air quality in China from 2013 to 2017, PNAS; Zhai et al., 2019. Fine particulate matter (PM2.5) trends in China, 2013–2018: separating contributions from anthropogenic emissions and meteorology, ACP; Zhong et al., 2018. Distinguishing Emission-Associated Ambient Air PM2.5 Concentrations and Meteorological Factor-Induced Fluctuations, EST. The authors should review the existing literature and put emphasize on their innovations.

Response: We thank the reviewer for the valuable suggestions on the introduction. We have supplemented the additional discussions to introduce more literature, including all of the above works as well as someone else, in order to substantiate and improve the "Introduction". Added/rewritten part in Sect. 1: The main challenge involves reliably representing substantial and rapid changes in anthropogenic emissions resulting from both long-term and emergency control measures (Chen et al., 2019; Cheng et al., 2019; Zhang et al., 2014; Yang et al., 2016; Zhai et al., 2019; Zhang et al., 2019; Zhong et al., 2018). To gain timely insight into variations in anthropogenic emissions, considerable efforts went into establishing detailed bottom-up emissions and derived valuable findings (Cheng et al., 2019; Zhang et al., 2019). Yet bottom-up inventories were built on the basis of activity data and emission factors. These input data can be absent or outdated, likely leading to misunderstandings of anthropogenic impacts, particularly in terms of the magnitude (Jiang et al., 2018). Recent studies applied available observations to construct multilinear regression models (emission-based or

meteorology-related), allowing us to separate contributions from anthropogenic emissions and meteorology to some extent (Zhai et al., 2019; Zhong et al., 2018). However, the uncertainties in bottom-up inventories and meteorological fields remained. Here we switched to observational constraints on a state-of-the-art chemical model. This can be a potential way to tackle this challenge.

2. I also have concerns about the methodology. Assimilation is used for calculating the total effect (emis+met) which gives a good representation of PM2.5 distributions, despite any model errors. But assimilation cannot be used for calculating met-only effect. Therefore, model errors may propagate into the met effect. I wonder what uncertainties this inconsistency in two pairs of simulations would cause for the results. The authors evaluated model emissions and concluded the impact is small. But it is not shown if other model errors may be significant. For example, studies have found that model tends to underestimate sulfate production during high RH in China. More evaluation of the model performance may be useful for interpreting the result.

Response: Thanks. Yes, we agree with the reviewer that more detailed model evaluation might be useful for this study. As previous studies have demonstrated (Cheng et al., 2019; Zhai et al., 2019; Zhong et al., 2018), model uncertainties remain, although we have verified the constrained results. We have supplemented the additional discussions in Sect. 4 for further explanations. For instance, model simulations of aerosol components (e.g., sulfate and nitrate) are still poorly constrained. Moreover, they have not been evaluated due to lack of available observations. Previous studies find that the model tends to underestimate sulfate production during high RH (as pointed by the reviewer) as well as SOA (Li et al., 2017a; Wang et al., 2014; Zhong et al., 2018). As a result, these uncertainties can be propagated into the estimations of meteorological effects. In addition, like other atmospheric chemical transport models, the WRF-CMAQ model cannot provide model uncertainty information in the simulations, while Monte Carlo simulations for complex CTMs would be unrealistic due to extremely high computation loadings (Zhong et al., 2018). Looking forward, more detailed model evaluations, as well as more explicit observational constraints, are of great significance for improving associated understandings, which will be the topic of a next separate study. Added/rewritten part in Sect. 4: Note that, as previous studies have demonstrated (Cheng et al., 2019; Zhai et al., 2019; Zhong et al., 2018), model uncertainties remain, although we have verified the constrained results. We have supplemented the additional discussions in Sect. 4 for further explanation. For instance, model simulations of aerosol components (e.g., sulfate and nitrate) are still poorly constrained. Moreover, they have not been evaluated due to lack of available observations. Yet previous studies find that the model tends to underestimate sulfate production during high RH and SOA (Li et al., 2017a; Wang et al., 2014; Zhong et al., 2018). As a result, these uncertainties can be propagated into the estimations of meteorological effects. Besides, like other atmospheric chemical transport models, the WRF-CMAQ model cannot provide model uncertainty information in the simulations, while Monte Carlo simulations for complex CTMs would be unrealistic due to extremely high computation loadings (Zhong et al., 2018).

3. The inclusion of short-term G20 period is interesting. But I am not completely convinced that the mitigation potential map is useful at all. At a first glance, the map does not seem to be very different from conducting a zero-ish YRD emission simulation with the model and then do a subtraction. The problem is that the authors did not provide information about (1) what types or fractions of emissions were shut down during the event; (2) is the emission shutdown implemented in Hangzhou, or Zhejiang, or YRD? Without this information, it is not possible to interpret the mitigation potential.

Response: Thanks. We have supplemented the information associated with anthropogenic emission control measures during the G20 summit. On that occasion, anthropogenic emission controls were conducted across the whole YRD (including Zhejiang, Jiangsu, and Anhui provinces, and Shanghai municipality), particularly in Hangzhou that served as the host city (Li et al., 2019, 2017b; Ni et al., 2020; Yu et al., 2018). Li et al. (2017) showed that most of anthropogenic emissions (e.g., those from industry, power plant, residential, and on-road transportation sectors) were reduced by around 50% on the basis of available governmental information. Added/rewritten part in Sect. 1: Those measures were conducted across the whole YRD (including Zhejiang, Jiangsu, and Anhui provinces, and Shanghai municipality), particularly in Hangzhou that served as the host city (Li et al., 2019, 2017b; Ni et al., 2020; Yu et al., 2018). Li et al. (2017) showed that most of anthropogenic emissions (e.g., those from industry, power plant, residential, and on-road transportation sectors) were reduced by around 50% on the basis of available governmental information.

Specific comments: 1. Line 41: Not clear from the text whether "> 14 $\mu$g/m3, 19%" is PM2.5 levels, or in fact, reduction in PM2.5 concentrations. Please clarify.

Response: Thanks. We have deleted the numbers. Specific numbers have been given in the following part of "Abstract".

2. Line 42-44: Confusing, as it interrupts the flow and misleads a reader that the decline in Hangzhou (35 ug/m3) is due to G20 control measures. I suggest moving the sentence to either Line 40 after "YRD, China" or to Line 48 before "Compared to the long-term policies...".

Response: Thanks. We suggest not to move this sentence. It follows behind the statement "For the winter time periods from 2016 to 2019" and is thus linked with the effects of the long-term policies from 2016 to 2019.

3. Line 46: remove "thus"

Response: Thanks. We have removed the word.

4. Line 99: should -> can

Response: Thanks. We have revised the word accordingly.

5. Line 105: "unprecedented" is a too-big word here that I suggest to remove. Same for other occurrences of the word in the paper.

Response: Thanks. We have removed the word throughout the paper.

6. Line 125: Be consistent with the citation format.

Response: Thanks. We have revised the format.

7. Line 137-139: Are meteorological observations assimilated in addition to chemical observations? If so, describe the meteorological observations that are assimilated. If not, I don't think it is sufficient to just use initial and boundary conditions from reanalysis data. The WRF should be run in a nudging mode, so the meteorology is close to reality.

Response: Thanks. Yes, the ECMWF reanalysis datasets were used to constrain meteorological simulations. Therein almost all necessary meteorological factors (nine variables), involving temperature, U wind component, V wind component, pressure, relative humidity, precipitation, short-wave radiation, cloud cover, and planetary boundary layer height (PBLH), were assimilated (https://apps.ecmwf.int/datasets/data/interim-full-daily/levtype=sfc/, last access: 7 March 2020). Added/rewritten part in Sect. 2.4: The ECMWF reanalysis datasets accounted for the hourly observational constraints on spatiotemporal meteorological evolutions. Therein almost all necessary meteorological factors (9 variables), involving temperature, U wind component, V wind component, pressure, relative humidity, precipitation, short-wave radiation, cloud cover, and planetary boundary layer height (PBLH), were assimilated (https://apps.ecmwf.int/datasets/data/interim-full-daily/levtype=sfc/, last access: 7 March 2020).

8. Line 155: "Prior anthropogenic emissions"? Do you optimize emissions at all? I cannot find such description throughout the text. If not, it should not be called "prior".

Response: Thanks. We have revised the word accordingly.

9. Line 166: "more" is not a proper conjunction word in formal English writing.

Response: Thanks. We have revised the word to "Moreover".

10. Line 176: grids -> grid cells

Response: Thanks. We have revised the word accordingly.

11. Line 176: "potential excellent roles" Rephrase it.

Response: Thanks. We have revised the phrase to "potentially excellent roles".

12. Section 2.4: What is the assimilation window? Daily? hourly?

Response: Thanks. The assimilation window is hourly. We have highlighted this in the Sect. 2.4. Added/rewritten part in Sect. 2.4: When ground-level PM2.5 measurements were assimilated, hourly observations were put into equation (1) to construct the new analysis fields. All-day state variables associated with aerosols in the model were adjusted from their background (simulated) to their analysis (constrained) states using the scaling factors ($X^a/X^b$).

13. Line 212: "the threshold pinpointing the key value of the correlation coefficients (e-1)". -> e-folding length

Response: Thanks. We have revised the sentence. Added/rewritten part in Sect. 2.3: The results indicated that a correlation length scale of âĹij 180 km could be treated as the threshold. It allowed the correlation coefficients to fall within the range of $e^{(-1)}$, defining the effective radius of each individual observation.

14. Line 217-219: well, it is still static in the relative sense. I don't this means anything. I'd suggest removing this statement.

Response: Thanks. We have removed the sentence.

15. Line 231: "unify the chemical inputs for the WRF-CMAQ model". What does this mean?

Response: Thanks. We have revised the sentence to further clarify the meaning. Added/rewritten part in Sect. 2.4: These configurations unified both chemical (i.e.,

emission inventories) and meteorological input data for the WRF-CMAQ model.

16. Line 237-240: Up to this point, we still do not know what method the authors use to separate effects of meteorology and emission. A clear description of the method is needed before this point.

Response: Thanks. We have revised the sentence to further clarify the meaning. Added/rewritten part in Sect. 2.4: Specifically, the differences in the constrained PM2.5 concentrations between DA_2016 and DA_2019 reflected the net effects of anthropogenic emission controls and meteorological perturbations between 2016 and 2019, while meteorological impacts therein were calculated as the differences in simulated PM2.5 concentrations between NO_2016 and NO_2019 (Chen et al., 2019a). Hence, by subtracting meteorological impacts from the net effects, we can isolate the effects of anthropogenic emission controls attributable to the long-term strategies.

17. Line 257: It is cursory to conclude these three periods have similar meteorology based on Fig. S1. The validity of the analysis is relied on the assumption that they are similar. E.g., one factor that is not analyzed is wind direction. Showing maps of circulation pattern will also help.

Response: Thanks. We have supplemented the map of atmospheric synoptic circulation patterns in Figure S1 accordingly (Dong et al., 2020; Liu et al., 2019).

18. Fig. 4. Is Fig. 4 useful? It is no surprising that the assimilated simulation could better reproduce observations, which are used in assimilation. It means nothing.

Response: Thanks. This figure is used to verify whether the model coupled with the OI method could reproduce the measurements. While 244 monitoring stations reside in 6660 grid cells, 16 grid cells have two to three monitors in them. For these grid cells, only one averaged measurement was used for DA. However, all the observations were compared against the constrained results. Hence, we suggest not to remove the figure and have supplemented the additional discussions in the Sect. 2.4. Added/rewritten

part in Sect. 2.4: While 244 monitoring stations reside in 6660 grid cells, 16 grid cells have two to three monitors in them. For these grid cells, only one averaged measurement was used for DA. However, all the observations were compared against the constrained results in the analyses.

19. Line 307-311. There is a jump in the logic of this sentence. I'd remove it.

Response: Thanks. We have removed the sentence.

20. Line 350-352: Many studies have properly separated the effects from meteorology and emissions, though with different approaches. I don't think the statement is fair.

Response: Thanks. We have revised the sentence. Added/rewritten part in Sect. 3.2: This also indirectly implied the importance of assimilating meteorology, which, however, were generally neglected by previous studies (Chen et al., 2019).

21. Line 353: It's unclear to me whether 5% and 3% are relative to mean PM2.5 concentration or mean reduction of PM2.5. Be more explicit.

Response: Thanks. We have added the absolute concentrations to make it clear. Added/rewritten part in Sect. 3.2: As shown in Figure S10 and Figure 5, even with the largest adjustment (i.e., -40%), such interferences could be well controlled within the 5% (< 3 $\mu$g/m3) scope, let alone other tests (i.e., < 3%, < 2 $\mu$g/m3).

22. Line 355: how do you prove it was "under the same meteorological condition".

Response: Thanks. We have revised the sentence to further clarify the meaning. Added/rewritten part in Sect. 3.2: Moreover, these findings are consistent with previous analyses (Cheng et al., 2019; Zhang et al., 2019). They generally revealed that reasonable changes in the bottom-up emissions, together with the same meteorology input data, would not remarkably alter the simulated results associated with meteorological effects on surface PM2.5 (< 5%).

23. Line 370: what is "concurrent meteorology"? better to rephrase it.

Response: Thanks. We have revised the phrase to "meteorological conditions therein".

24. Line 375-376: Of course, the long-term strategies are emission oriented. You cannot change weather easily after all ... I guess the author wanted to say the long-term decrease in PM2.5 was driven mainly by decreased emissions.

Response: Thanks. We agree. We have revised the sentence. Added/rewritten part in Sect. 3.2: This indicates that the impacts of the long-term strategies are mainly driven by anthropogenic emission mitigation.

25. Line 387: what is "stably spatiotemporal state". Rephrase it.

Response: Thanks. We have revised the sentence. Added/rewritten part in Sect. 3.3: We found that such impacts were of relatively low standard deviations (< 5%) and kept stable over time.

26. Line 412: remove "unprecedented".

Response: Thanks. We have removed the word throughout the paper.

27. Line 414: what is "stable supply-side structures"? Not directly related to air quality to me.

Response: Thanks. We have revised the phase to "stable structures of anthropogenic emissions".

28. Line 418: There have been quite a few papers discussing the effect of emission control strategy. To list a few, Zhang et al., 2019. Drivers of improved PM2.5 air quality in China from 2013 to 2017, PNAS Zhai et al., 2019. Fine particulate matter (PM2.5) trends in China, 2013–2018: separating contributions from anthropogenic emissions and meteorology, ACP Zhong et al., 2018. Distinguishing Emission-Associated Ambient Air PM2.5 Concentrations and Meteorological Factor-Induced Fluctuations, EST Although the method may not be the same, the authors should give credit to these studies rather than claim this is the first study trying to separate effects of met and

emission.

Response: Thanks. We have addressed this issue in the response for the general comment (1). We have supplemented the additional discussions to introduce more literature, including all of the above works as well as someone else, in order to substantiate and improve the "Introduction". Added/rewritten part in Sect. 1: The main challenge involves reliably representing substantial and rapid changes in anthropogenic emissions resulting from both long-term and emergency control measures (Chen et al., 2019; Cheng et al., 2019; Zhang et al., 2014; Yang et al., 2016; Zhai et al., 2019; Zhang et al., 2019; Zhong et al., 2018). To gain timely insight into variations in anthropogenic emissions, considerable efforts went into establishing detailed bottom-up emissions and derived valuable findings (Cheng et al., 2019; Zhang et al., 2019). Yet bottom-up inventories were built on the basis of activity data and emission factors. These input data can be absent or outdated, likely leading to misunderstandings of anthropogenic impacts, particularly in terms of the magnitude (Jiang et al., 2018). Recent studies applied available observations to construct multilinear regression models (emission-based or meteorology-related), allowing us to separate contributions from anthropogenic emissions and meteorology to some extent (Zhai et al., 2019; Zhong et al., 2018). However, the uncertainties in bottom-up inventories and meteorological fields remained. Here we switched to observational constraints on a state-of-the-art chemical model. This can be a potential way to tackle this challenge.

29. Line 444: "rudimentary" may not be a proper word here.

Response: Thanks. We have removed the word.

30. Line 449: the statement that "the biogenic emissions are unimportant for IAV of PM2.5" may be true for YRD, but may not be "generally" true for elsewhere in the world. I'd suggest being more specific.

Response: Thanks. We have revised the sentence to make it more specific. Added/rewritten part in Sect. 4: Moreover, the former is generally of minor significance

for interannual PM2.5 variations for the YRD.

References: Chen, D., Liu, Z., Ban, J., Zhao, P. and Chen, M.: Retrospective analysis of 2015–2017 wintertime PM 2.5 in China: response to emission regulations and the role of meteorology, Atmos. Chem. Phys., 19(11), 7409–7427, 2019a.

Chen, D., Liu, Z., Ban, J., Zhao, P. and Chen, M.: Retrospective analysis of 2015-2017 wintertime PM2.5 in China: response to emission regulations and the role of meteorology, Atmos. Chem. Phys., 19(11), 7409–7427, 2019b.

Chen, D., Liu, Z., Ban, J. and Chen, M.: The 2015 and 2016 wintertime air pollution in China: SO2 emission changes derived from a WRF-Chem/EnKF coupled data assimilation system, Atmos. Chem. Phys., 19(13), 8619–8650, 2019c.

Cheng, J., Su, J., Cui, T., Li, X., Dong, X., Sun, F., Yang, Y., Tong, D., Zheng, Y., Li, Y. and others: Dominant role of emission reduction in PM2.5 air quality improvement in Beijing during 2013–2017: A model-based decomposition analysis, Atmos. Chem. Phys., 19(9), 6125–6146, 2019.

Dong, Y., Li, J., Guo, J., Jiang, Z., Chu, Y., Chang, L., Yang, Y. and Liao, H.: The impact of synoptic patterns on summertime ozone pollution in the North China Plain, Sci. Total Environ., 735, 139559, doi:https://doi.org/10.1016/j.scitotenv.2020.139559, 2020.

Jiang, Z., McDonald, B. C., Worden, H., Worden, J. R., Miyazaki, K., Qu, Z., Henze, D. K., Jones, D. B. A., Arellano, A. F., Fischer, E. V and others: Unexpected slowdown of US pollutant emission reduction in the past decade, Proc. Natl. Acad. Sci., 115(20), 5099–5104, 2018.

Li, G., Bei, N., Cao, J., Huang, R., Wu, J., Feng, T., Wang, Y., Liu, S., Zhang, Q., Tie, X. and Molina, L. T.: A possible pathway for rapid growth of sulfate during haze days in China, Atmos. Chem. Phys., 17(5), 3301–3316, doi:10.5194/acp-17-3301-2017, 2017a.

Li, H., Wang, D., Cui, L., Gao, Y., Huo, J., Wang, X., Zhang, Z., Tan, Y., Huang, Y.,

Cao, J. and others: Characteristics of atmospheric PM2. 5 composition during the implementation of stringent pollution control measures in shanghai for the 2016 G20 summit, Sci. Total Environ., 648, 1121–1129, 2019.

Li, P., Wang, L., Guo, P., Yu, S., Mehmood, K., Wang, S., Liu, W., Seinfeld, J. H., Zhang, Y., Wong, D. C. and others: High reduction of ozone and particulate matter during the 2016 G-20 summit in Hangzhou by forced emission controls of industry and traffic, Environ. Chem. Lett., 15(4), 709–715, 2017.

Liu, N., Zhou, S., Liu, C. and Guo, J.: Synoptic circulation pattern and boundary layer structure associated with PM2.5 during wintertime haze pollution episodes in Shanghai, Atmos. Res., 228, 186–195, doi:https://doi.org/10.1016/j.atmosres.2019.06.001, 2019.

Ni, Z.-Z., Luo, K., Gao, Y., Gao, X., Jiang, F., Huang, C., Fan, J.-R., Fu, J. S. and Chen, C.-H.: Spatial–temporal variations and process analysis of O3 pollution in Hangzhou during the G20 summit, Atmos. Chem. Phys., 20(10), 5963–5976, doi:10.5194/acp-20-5963-2020, 2020.

Wang, Y., Zhang, Q., Jiang, J., Zhou, W., Wang, B., He, K., Duan, F., Zhang, Q., Philip, S. and Xie, Y.: Enhanced sulfate formation during China's severe winter haze episode in January 2013 missing from current models, J. Geophys. Res. Atmos., 119(17), 10,410-425,440, doi:10.1002/2013JD021426, 2014.

Yang, Y., Liao, H. and Lou, S.: Increase in winter haze over eastern China in recent decades: Roles of variations in meteorological parameters and anthropogenic emissions, J. Geophys. Res. Atmos., 121(21), 13,13-50,65, doi:10.1002/2016JD025136, 2016.

Yu, H., Dai, W., Ren, L., Liu, D., Yan, X., Xiao, H., He, J. and Xu, H.: The effect of emission control on the submicron particulate matter size distribution in Hangzhou during the 2016 G20 Summit, Aerosol Air Qual. Res., 18, 2038–2046, 2018.

[Figure]

Zhai, S., Jacob, D. J., Wang, X., Shen, L., Li, K., Zhang, Y., Gui, K., Zhao, T. and Liao, H.: Fine particulate matter (PM2.5) trends in China, 2013–2018: separating contributions from anthropogenic emissions and meteorology, Atmos. Chem. Phys., 19(16), 11031–11041, doi:10.5194/acp-19-11031-2019, 2019.

Zhang, R., Li, Q. and Zhang, R.: Meteorological conditions for the persistent severe fog and haze event over eastern China in January 2013, Sci. China Earth Sci., 57(1), 26–35, doi:10.1007/s11430-013-4774-3, 2014.

Zhang, Q., Zheng, Y., Tong, D., Shao, M., Wang, S., Zhang, Y., Xu, X., Wang, J., He, H., Liu, W. and others: Drivers of improved PM2.5 air quality in China from 2013 to 2017, Proc. Natl. Acad. Sci., 116, 24463–24469,2019.

Zhong, Q., Ma, J., Shen, G., Shen, H., Zhu, X., Yun, X., Meng, W., Cheng, H., Liu, J., Li, B., Wang, X., Zeng, E. Y., Guan, D. and Tao, S.: Distinguishing emission-associated ambient air PM2.5 concentrations and meteorological factor-induced fluctuations, Environ. Sci. Technol., 52(18), 10416–10425, doi:10.1021/acs.est.8b02685, 2018.
* * *

---

## Author Response (AR1)

5

Anonymous Referee #1 Received and published: 20 August 2020

**General comments:**

- 10 Wang et al. evaluated the effect of long-term and emergent emission control strategies on the PM2.5 levels in Yangtze River Delta of China, by combining modelling analysis with observations. They found the decline in PM2.5 concentration during 2016-2019 was mainly due to emission control. The decline would be even greater if the meteorology was not unfavorable. Great potential of further decrease is manifested in analysis of data during G20 period when short-term emergent measures were taken. The discussion is valuable for assessment of past policies and design of future ones. However, to be publishable
- 15 in ACP, the current manuscript requires further improvement.

**Response:** We thank the reviewer for the thoughtful comments on our paper and have addressed them as below.

 Particularly, inadequate credits are given to the existing literature that performed similar analysis of separating meteorology and emission effects on recent PM2.5 trend in China. Instead, the authors tried to impress the reader by suggesting that this
 study is the first to do so. Just to list a few studies in literature (and I believe there might be more), Zhang et al., 2019. Drivers of improved PM2.5 air quality in China from 2013 to 2017, PNAS; Zhai et al., 2019. Fine particulate matter (PM2.5) trends in China, 2013–2018: separating contributions from anthropogenic emissions and meteorology, ACP; Zhong et al., 2018. Distinguishing Emission-Associated Ambient Air PM2.5 Concentrations and Meteorological Factor-Induced Fluctuations, EST. The authors should review the existing literature and put emphasize on their innovations.

25 **Response:** We thank the reviewer for the valuable suggestions on the introduction. We have supplemented the additional discussions to introduce more literature, including all of the above works as well as someone else, in order to substantiate and improve the "Introduction".

Added/rewritten part in Sect. 1: The main challenge involves reliably representing substantial and rapid changes in anthropogenic emissions resulting from both long-term and emergency control measures (Chen et al., 2019; Cheng et al., 2019;

30 Zhang et al., 2014; Yang et al., 2016; Zhai et al., 2019; Zhang et al., 2019; Zhong et al., 2018). To gain timely insight into variations in anthropogenic emissions, considerable efforts went into establishing detailed bottom-up emissions and derived valuable findings (Cheng et al., 2019; Zhang et al., 2019). Yet bottom-up inventories were built on the basis of activity data

and emission factors. These input data can be absent or outdated, likely leading to misunderstandings of anthropogenic impacts, particularly in terms of the magnitude (Jiang et al., 2018). Recent studies applied available observations to construct multilinear

- 35 regression models (emission-based or meteorology-related), allowing us to separate contributions from anthropogenic emissions and meteorology to some extent (Zhai et al., 2019; Zhong et al., 2018). However, the uncertainties in bottom-up inventories and meteorological fields remained. Here we switched to observational constraints on a state-of-the-art chemical model. This can be a potential way to tackle this challenge.
- 40 2. I also have concerns about the methodology. Assimilation is used for calculating the total effect (emis+met) which gives a good representation of PM2.5 distributions, despite any model errors. But assimilation cannot be used for calculating met-only effect. Therefore, model errors may propagate into the met effect. I wonder what uncertainties this inconsistency in two pairs of simulations would cause for the results. The authors evaluated model emissions and concluded the impact is small. But it is not shown if other model errors may be significant. For example, studies have found that model tends to underestimate sulfate production during high RH in China. More evaluation of the model performance may be useful for interpreting the result.
- **Response:** Thanks. Yes, we agree with the reviewer that more detailed model evaluation might be useful for this study. As previous studies have demonstrated (Cheng et al., 2019; Zhai et al., 2019; Zhong et al., 2018), model uncertainties remain, although we have verified the constrained results. We have supplemented the additional discussions in Sect. 4 for further explanations. For instance, model simulations of aerosol components (e.g., sulfate and nitrate) are still poorly constrained.
- 50 Moreover, they have not been evaluated due to lack of available observations. Previous studies find that the model tends to underestimate sulfate production during high RH (as pointed by the reviewer) as well as SOA (Li et al., 2017a; Wang et al., 2014; Zhong et al., 2018). As a result, these uncertainties can be propagated into the estimations of meteorological effects. In addition, like other atmospheric chemical transport models, the WRF-CMAQ model cannot provide model uncertainty information in the simulations, while Monte Carlo simulations for complex CTMs would be unrealistic due to extremely high
- 55 computation loadings (Zhong et al., 2018). Looking forward, more detailed model evaluations, as well as more explicit observational constraints, are of great significance for improving associated understandings, which will be the topic of a next separate study.

Added/rewritten part in Sect. 4: Note that, as previous studies have demonstrated (Cheng et al., 2019; Zhai et al., 2019; Zhong et al., 2018), model uncertainties remain, although we have verified the constrained results. We have supplemented the

- 60 additional discussions in Sect. 4 for further explanation. For instance, model simulations of aerosol components (e.g., sulfate and nitrate) are still poorly constrained. Moreover, they have not been evaluated due to lack of available observations. Yet previous studies find that the model tends to underestimate sulfate production during high RH and SOA (Li et al., 2017a; Wang et al., 2014; Zhong et al., 2018). As a result, these uncertainties can be propagated into the estimations of meteorological effects. Besides, like other atmospheric chemical transport models, the WRF-CMAQ model cannot provide model uncertainty
- 65 information in the simulations, while Monte Carlo simulations for complex CTMs would be unrealistic due to extremely high computation loadings (Zhong et al., 2018).

3. The inclusion of short-term G20 period is interesting. But I am not completely convinced that the mitigation potential map is useful at all. At a first glance, the map does not seem to be very different from conducting a zero-ish YRD emission

70 simulation with the model and then do a subtraction. The problem is that the authors did not provide information about (1) what types or fractions of emissions were shut down during the event; (2) is the emission shutdown implemented in Hangzhou, or Zhejiang, or YRD? Without this information, it is not possible to interpret the mitigation potential.

**Response:** Thanks. We have supplemented the information associated with anthropogenic emission control measures during the G20 summit. On that occasion, anthropogenic emission controls were conducted across the whole YRD (including Zhejiang,

75 Jiangsu, and Anhui provinces, and Shanghai municipality), particularly in Hangzhou that served as the host city (Li et al., 2019, 2017b; Ni et al., 2020; Yu et al., 2018). Li et al. (2017) showed that most of anthropogenic emissions (e.g., those from industry, power plant, residential, and on-road transportation sectors) were reduced by around 50% on the basis of available governmental information.

Added/rewritten part in Sect. 1: Those measures were conducted across the whole YRD (including Zhejiang, Jiangsu, and

80 Anhui provinces, and Shanghai municipality), particularly in Hangzhou that served as the host city (Li et al., 2019, 2017b; Ni et al., 2020; Yu et al., 2018). Li et al. (2017) showed that most of anthropogenic emissions (e.g., those from industry, power plant, residential, and on-road transportation sectors) were reduced by around 50% on the basis of available governmental information.

**85 Specific comments:**

1. Line 41: Not clear from the text whether "> 14  $\mu$ g/m3, 19%" is PM2.5 levels, or in fact, reduction in PM2.5 concentrations. Please clarify.

Response: Thanks. We have deleted the numbers. Specific numbers have been given in the following part of "Abstract".

90 2. Line 42-44: Confusing, as it interrupts the flow and misleads a reader that the decline in Hangzhou (35 ug/m3) is due to G20 control measures. I suggest moving the sentence to either Line 40 after "YRD, China" or to Line 48 before "Compared to the long-term policies...".

**Response: Thanks.** We suggest not to move this sentence. It follows behind the statement "For the winter time periods from 2016 to 2019" and is thus linked with the effects of the long-term policies from 2016 to 2019.

**95**

3. Line 46: remove "thus"

**Response:** Thanks. We have removed the word.

4. Line 99: should -> can

**Response:** Thanks. We have revised the word accordingly.

5. Line 105: "unprecedented" is a too-big word here that I suggest to remove. Same for other occurrences of the word in the paper.

Response: Thanks. We have removed the word throughout the paper.

105 6. Line 125: Be consistent with the citation format.

Response: Thanks. We have revised the format.

7. Line 137-139: Are meteorological observations assimilated in addition to chemical observations? If so, describe the meteorological observations that are assimilated. If not, I don't think it is sufficient to just use initial and boundary conditions from reanalysis data. The WRF should be run in a nudging mode, so the meteorology is close to reality.

- **Response:** Thanks. Yes, the ECMWF reanalysis datasets were used to constrain meteorological simulations. Therein almost all necessary meteorological factors (nine variables), involving temperature, U wind component, V wind component, pressure, relative humidity, precipitation, short-wave radiation, cloud cover, and planetary boundary layer height (PBLH), were assimilated (https://apps.ecmwf.int/datasets/data/interim-full-daily/levtype=sfc/, last access: 7 March 2020).
- 115 Added/rewritten part in Sect. 2.4: The ECMWF reanalysis datasets accounted for the hourly observational constraints on spatiotemporal meteorological evolutions. Therein almost all necessary meteorological factors (9 variables), involving temperature, U wind component, V wind component, pressure, relative humidity, precipitation, short-wave radiation, cloud cover, and planetary boundary layer height (PBLH), were assimilated (https://apps.ecmwf.int/datasets/data/interim-full-daily/levtype=sfc/, last access: 7 March 2020).

120

110

8. Line 155: "Prior anthropogenic emissions"? Do you optimize emissions at all? I cannot find such description throughout the text. If not, it should not be called "prior".

**Response:** Thanks. We have revised the word accordingly.

125 9. Line 166: "more" is not a proper conjunction word in formal English writing.Response: Thanks. We have revised the word to "Moreover".

10. Line 176: grids -> grid cells

**Response:** Thanks. We have revised the word accordingly.

130

11. Line 176: "potential excellent roles" Rephrase it.**Response:** Thanks. We have revised the phrase to "potentially excellent roles".

12. Section 2.4: What is the assimilation window? Daily? hourly?

- 135 Response: Thanks. The assimilation window is hourly. We have highlighted this in the Sect. 2.4. Added/rewritten part in Sect. 2.4: When ground-level PM2.5 measurements were assimilated, hourly observations were put into equation (1) to construct the new analysis fields. All-day state variables associated with aerosols in the model were adjusted from their background (simulated) to their analysis (constrained) states using the scaling factors (Xa/Xb).
- 13. Line 212: "the threshold pinpointing the key value of the correlation coefficients (e-1)". -> e-folding length
  Response: Thanks. We have revised the sentence.
  Added/rewritten part in Sect. 2.3: The results indicated that a correlation length scale of ~ 180 km could be treated as the threshold. It allowed the correlation coefficients to fall within the range of e-1, defining the effective radius of each individual observation.

155

165

14. Line 217-219: well, it is still static in the relative sense. I don't this means anything. I'd suggest removing this statement. **Response:** Thanks. We have removed the sentence.

15. Line 231: "unify the chemical inputs for the WRF-CMAQ model". What does this mean?

Response: Thanks. We have revised the sentence to further clarify the meaning.
 Added/rewritten part in Sect. 2.4: These configurations unified both chemical (i.e., emission inventories) and meteorological input data for the WRF-CMAQ model.

16. Line 237-240: Up to this point, we still do not know what method the authors use to separate effects of meteorology and emission. A clear description of the method is needed before this point.

**Response:** Thanks. We have revised the sentence to further clarify the meaning.

Added/rewritten part in Sect. 2.4: Specifically, the differences in the constrained  $PM_{2.5}$  concentrations between DA\_2016 and DA\_2019 reflected the net effects of anthropogenic emission controls and meteorological perturbations between 2016 and 2019, while meteorological impacts therein were calculated as the differences in simulated  $PM_{2.5}$  concentrations between

160 NO\_2016 and NO\_2019 (Chen et al., 2019a). Hence, by subtracting meteorological impacts from the net effects, we can isolate the effects of anthropogenic emission controls attributable to the long-term strategies.

17. Line 257: It is cursory to conclude these three periods have similar meteorology based on Fig. S1. The validity of the analysis is relied on the assumption that they are similar. E.g., one factor that is not analyzed is wind direction. Showing maps of circulation pattern will also help.

**Response:** Thanks. We have supplemented the map of atmospheric synoptic circulation patterns in Figure S1 accordingly (Dong et al., 2020; Liu et al., 2019).

18. Fig. 4. Is Fig. 4 useful? It is no surprising that the assimilated simulation could better reproduce observations, which are used in assimilation. It means nothing.

**Response:** Thanks. This figure is used to verify whether the model coupled with the OI method could reproduce the measurements. While 244 monitoring stations reside in 6660 grid cells, 16 grid cells have two to three monitors in them. For these grid cells, only one averaged measurement was used for DA. However, all the observations were compared against the constrained results. Hence, we suggest not to remove the figure and have supplemented the additional discussions in the Sect.

175 2.4.

170

Added/rewritten part in Sect. 2.4: While 244 monitoring stations reside in 6660 grid cells, 16 grid cells have two to three monitors in them. For these grid cells, only one averaged measurement was used for DA. However, all the observations were compared against the constrained results in the analyses.

180 19. Line 307-311. There is a jump in the logic of this sentence. I'd remove it.Response: Thanks. We have removed the sentence.

20. Line 350-352: Many studies have properly separated the effects from meteorology and emissions, though with different approaches. I don't think the statement is fair.

185 **Response:** Thanks. We have revised the sentence.

Added/rewritten part in Sect. 3.2: This also indirectly implied the importance of assimilating meteorology, which, however, were generally neglected by previous studies (Chen et al., 2019).

21. Line 353: It's unclear to me whether 5% and 3% are relative to mean  $PM_{2.5}$  concentration or mean reduction of  $PM_{2.5}$ . Be 190 more explicit.

Response: Thanks. We have added the absolute concentrations to make it clear.

Added/rewritten part in Sect. 3.2: As shown in Figure S10 and Figure 5, even with the largest adjustment (i.e., -40%), such interferences could be well controlled within the 5% ( $< 3 \mu g/m^3$ ) scope, let alone other tests (i.e., < 3%,  $< 2 \mu g/m^3$ ).

195 22. Line 355: how do you prove it was "under the same meteorological condition".

**Response:** Thanks. We have revised the sentence to further clarify the meaning.

Added/rewritten part in Sect. 3.2: Moreover, these findings are consistent with previous analyses (Cheng et al., 2019; Zhang et al., 2019). They generally revealed that reasonable changes in the bottom-up emissions, together with the same meteorology input data, would not remarkably alter the simulated results associated with meteorological effects on surface  $PM_{2.5}$  (< 5%).

200

23. Line 370: what is "concurrent meteorology"? better to rephrase it.

Response: Thanks. We have revised the phrase to "meteorological conditions therein".

24. Line 375-376: Of course, the long-term strategies are emission oriented. You cannot change weather easily after all ... I

205 guess the author wanted to say the long-term decrease in PM2.5 was driven mainly by decreased emissions.
 Response: Thanks. We agree. We have revised the sentence.
 Added/rewritten part in Sect. 3.2: This indicates that the impacts of the long-term strategies are mainly driven by anthropogenic emission mitigation.

- 210 25. Line 387: what is "stably spatiotemporal state". Rephrase it.
  Response: Thanks. We have revised the sentence.
  Added/rewritten part in Sect. 3.3: We found that such impacts were of relatively low standard deviations (< 5%) and kept stable over time.</li>
- 215 26. Line 412: remove "unprecedented".

**Response:** Thanks. We have removed the word throughout the paper.

27. Line 414: what is "stable supply-side structures"? Not directly related to air quality to me.

Response: Thanks. We have revised the phase to "stable structures of anthropogenic emissions".

220

28. Line 418: There have been quite a few papers discussing the effect of emission control strategy. To list a few, Zhang et al., 2019. Drivers of improved  $PM_{2.5}$  air quality in China from 2013 to 2017, PNAS Zhai et al., 2019. Fine particulate matter ( $PM_{2.5}$ ) trends in China, 2013–2018: separating contributions from anthropogenic emissions and meteorology, ACP Zhong et al., 2018. Distinguishing Emission-Associated Ambient Air  $PM_{2.5}$  Concentrations and Meteorological Factor-Induced

225 Fluctuations, EST Although the method may not be the same, the authors should give credit to these studies rather than claim this is the first study trying to separate effects of met and emission.

**Response:** Thanks. We have addressed this issue in the response for the general comment (1). We have supplemented the additional discussions to introduce more literature, including all of the above works as well as someone else, in order to substantiate and improve the "Introduction".

- 230 Added/rewritten part in Sect. 1: The main challenge involves reliably representing substantial and rapid changes in anthropogenic emissions resulting from both long-term and emergency control measures (Chen et al., 2019; Cheng et al., 2019; Zhang et al., 2014; Yang et al., 2016; Zhai et al., 2019; Zhang et al., 2019; Zhong et al., 2018). To gain timely insight into variations in anthropogenic emissions, considerable efforts went into establishing detailed bottom-up emissions and derived valuable findings (Cheng et al., 2019; Zhang et al., 2019). Yet bottom-up inventories were built on the basis of activity data
- 235 and emission factors. These input data can be absent or outdated, likely leading to misunderstandings of anthropogenic impacts, particularly in terms of the magnitude (Jiang et al., 2018). Recent studies applied available observations to construct multilinear

regression models (emission-based or meteorology-related), allowing us to separate contributions from anthropogenic emissions and meteorology to some extent (Zhai et al., 2019; Zhong et al., 2018). However, the uncertainties in bottom-up inventories and meteorological fields remained. Here we switched to observational constraints on a state-of-the-art chemical model. This can be a potential way to tackle this challenge.

240

29. Line 444: "rudimentary" may not be a proper word here.

**Response:** Thanks. We have removed the word.

**245**

250

30. Line 449: the statement that "the biogenic emissions are unimportant for IAV of PM2.5" may be true for YRD, but may not be "generally" true for elsewhere in the world. I'd suggest being more specific.

**Response:** Thanks. We have revised the sentence to make it more specific.

Added/rewritten part in Sect. 4: Moreover, the former is generally of minor significance for interannual PM2.5 variations for the YRD.

20 on the YRD region. Second, how is the modelled PM2.5 constrained spatiotemporally by observations, applying DA generated scaling factors to the whole fan-shaped quadrilateral region, the YRD region, city by city, or grid by grid, and hour by hour or day by day?

**Response:** Thanks. We have supplemented the additional discussions in Sect. 2.3 to explain why we choose the ground-level observations within the fan-shaped quadrilateral to constrain the model performance. As pointed by the reviewer, we aim to

- 25 minimize the impacts outside the YRD region. Specifically, this was mainly due to the fact that this fan-shaped geographical scope covered almost all key regions that had potentially regional impacts on the YRD, involving the Beijing-Tianjin-Hebei region (BTH), the Pearl River Delta region, the Sichuan-Chongqing region, and the Shaanxi-Gansu region (Zhang et al., 2019). On the other hand, the ground monitoring sites within the fan-shaped quadrilateral were significantly denser than those outside, thus leading to much more effective DA in practice (Bocquet et al., 2015; Chai et al., 2017). Therefore, to assimilate the
- 30 observations within the fan-shaped quadrilateral might be a sensible way to balance the DA effectiveness and computing efficiency. A resultant evidence lies in the model performance evaluation in Sect. 3.1, which would prove that this DA

configuration can enable reliable  $PM_{2.5}$  simulations. Collectively, we might eliminate the need of the associated sensitivity analysis.

In addition, we have supplemented the more discussions in Sect. 2.4 to further detail how to conduct observational constraints

- on the model simulations spatiotemporally. In short, we conducted hourly DA for grid cells. Note that the effective radius of each individual observation should be calculated in advance. When ground-level PM2.5 measurements were assimilated, hourly observations were put into equation (1) to construct the new analysis fields. All-day state variables associated with aerosols in the model were adjusted from their background (simulated) to their analysis (constrained) states using the scaling factors  $(X^a/X^b)$ . The adjusted model state variables were then used to initiate the model to predict the next background state  $(X^b)$  in Equation (1). Therefore, the background state  $(X^b)$  served as a prior model prediction before it was combined with the newly
- 40 Equation (1). Therefore, the background state  $(\mathbf{X}^a)$  served as a prior model prediction before it was combined with the newly available observation (**Y**) to generate a new analysis state  $(\mathbf{X}^a)$  using Equation (1). Measurements within the background-error correlation length scale were used to shape analysis states  $(\mathbf{X}^a)$ . The background error covariance **COV**ii between any two grid cells **i** and **j** was simulated as

$$\mathbf{COV}_{ij} = \varepsilon_i \varepsilon_j \mathbf{e}^{-\frac{\Delta_{ij}}{L}}$$
(2)

- 45 where  $\mathbf{\varepsilon}_i$  and  $\mathbf{\varepsilon}_j$  referred to the standard deviations of the background errors in two grid cells and  $\Delta_{ij}$  denoted the distance between the two grids. As a result, **L** was the background-error correlation length scale, which can be the Hollingsworth-Lönnberg method (Chai et al., 2017; Hollingsworth and Lönnberg, 1986; Kumar et al., 2012). Figure 2 shows the correlation coefficient, i.e.,  $\mathbf{COV}_{ij}/\mathbf{\varepsilon}_i\mathbf{\varepsilon}_j$ , as a function of the separation distance between two grid cells, which was averaged over 10 km bins. The results indicate that a correlation length scale of ~ 180 km could be treated as the threshold by allowing the
- 50 correlation coefficients to fall within the range of  $e^{-1}$ , defining the effective radius of each individual observation. Due to the intensive monitoring sites in our study domain, this threshold was applied uniformly for the YRD. In this study, observations beyond the background-error correlation length scale would have no effect on  $X^a$ .

Added/rewritten part in Sect. 2.3: As shown in Figure 1a, to consider regional impacts outside the YRD, the ground-level observations in the fan-shaped quadrilateral were used to constrain the model performance. This was mainly due to the fact

- 55 that this fan-shaped geographical scope covered almost all key regions that had potentially regional impacts on the YRD, involving the Beijing-Tianjin-Hebei region (BTH), the Pearl River Delta region, the Sichuan-Chongqing region, and the Shaanxi-Gansu region (Zhang et al., 2019). On the other hand, the ground monitoring sites within the fan-shaped quadrilateral were significantly denser than those outside, thus leading to much more effective DA results in practice (Bocquet et al., 2015; Chai et al., 2017). Collectively, to assimilate the observations in the fan-shaped quadrilateral might be a sensible way to balance
- 60 the DA effectiveness and the computing efficiency. A resultant evidence lies in the model performance evaluation in Sect. 3.1, which would prove that this DA configuration can enable reliable PM2.5 simulations.

Added/rewritten part in Sect. 2.4: When ground-level  $PM_{2.5}$  measurements were assimilated, hourly observations were put into equation (1) to construct the new analysis fields. All-day state variables associated with aerosols in the model were adjusted from their background (simulated) to their analysis (constrained) states using the scaling factors ( $X^a/X^b$ ). The 65 adjusted model state variables were then used to initiate the model to predict the next background state  $(\mathbf{X}^b)$  in equation (1). Therefore, the background state  $(\mathbf{X}^b)$  served as a prior model prediction before it was combined with the newly available observation (**Y**) to generate a new analysis state ( $\mathbf{X}^a$ ) using Equation (1).

Measurements within the background-error correlation length scale were used to shape analysis states ( $X^a$ ). The background error covariance **COV**ii between any two grid cells **i** and **j** was simulated as

70  $\mathbf{COV}_{ii} = \boldsymbol{\varepsilon}_{i} \boldsymbol{\varepsilon}_{i} \mathbf{e}^{-\frac{\Delta_{ij}}{L}}$ (2)

where  $\varepsilon_i$  and  $\varepsilon_j$  referred to the standard deviations of the background errors in two grid cells and  $\Delta_{ij}$  denoted the distance between the two grids. As a result, **L** was the background-error correlation length scale, which can be obtained by the Hollingsworth-Lönnberg method (Chai et al., 2017; Hollingsworth and Lönnberg, 1986; Kumar et al., 2012). Figure 2 shows the correlation coefficient, i.e., **COV**ij/ $\varepsilon_i \varepsilon_j$ , as a function of the separation distance between two grid cells, which was averaged

- 75 over 10 km bins. The results indicated that a correlation length scale of ~ 180 km could be treated as the threshold allowing the correlation coefficients to fall within the range of  $e^{-1}$ , defining the effective radius of each individual observation. Due to the intensive monitoring sites in our study domain, this threshold was applied uniformly for the YRD. In this study, observations beyond the background-error correlation length scale were assumed to have no effect on  $X^a$ .
- 2. The author used a statistical method to establish the correlation coefficients and chose separation distance of 180 km as a threshold. The author needs to give more explanations on the value of chosen. If the purpose is to find a correlation length scale to minimize the effect on  $X^a$ , based on Fig 2, it seems that separation distance of 600 km would be more appropriate. **Response:** Thanks. The objective of identifying the background-error correlation length scale is to define the effective radius
- of each individual observation and thus to establish reliable analysis states ( $X^a$ ). Here the Hollingsworth-Lönnberg approach, wildly used for decades (Chai et al., 2017; Hollingsworth and Lönnberg, 1986; Kumar et al., 2012), is applied to calculate the background-error correlation length scale. Observations beyond the background-error correlation length scale were assumed to have no effect on  $X^a$ . Once observations far away are introduced, more background errors  $COV_{ij}$ , larger than  $e^{-1}$ , would be put into  $X^a$  as calculated in Equation (2). Corresponding detailed information has been given in the response for the specific comment (2).
- 90

3. How did the author isolate the impact from emission reductions on  $PM_{2.5}$  concentrations? Did the author use the constrained  $PM_{2.5}$  subtract the impact on simulated  $PM_{2.5}$  from meteorological variations? Even the modelled temperature, humidity, wind speed, and air pressure were also assimilated in this study, there are other parameters, for example, modelled PBL height, causing large uncertainties in the modelled meteorological field, and thus leading to bias and error in the calculated net impacts

95 from emission variations. For example, figures c and f in Fig 5, show very small impact of anthropogenic emission control from 2016 to 2019 in most of Zhejiang province compared to the other provinces in the YRD region. Is it reasonable?

**Response:** Thanks. Yes, it is reasonable. We isolated anthropogenic impacts on  $PM_{2.5}$  concentrations by subtracting the corresponding meteorological impacts from the constrained  $PM_{2.5}$  fields. To further illustrate the process of meteorological assimilations, we have supplemented the additional discussions in Sect. 2.4. The ECMWF reanalysis datasets accounted for

- 100 the hourly observational constraints on spatiotemporal meteorological evolutions. Therein almost all necessary meteorological factors (nine variables), involving temperature, U wind component, V wind component, pressure, relative humidity, precipitation, short-wave radiation, cloud cover, and boundary layer height, were assimilated (https://apps.ecmwf.int/datasets/data/interim-full-daily/levtype=sfc/, last access: 7 March 2020).
- The model evaluation provides a more direct way to verify the corresponding model performance. As highlighted in Sect. 3.1, 105 given the fact that the assimilated ERA reanalysis dataset has much wider spatial coverage than ground-based measurements, we also reproduced the spatiotemporal variations in the meteorological factors (e.g., temperature, relative humidity, wind speed, and air pressure) (Figures S5 ~ S8). Together with the comprehensive evaluation statistics as summarized in Tables S1 ~ S5, it has been demonstrated that the DA method can enable one to derive not only reliable PM2.5 evolution but also accurate meteorological fields.
- 110 In terms of the issue associated with Zhejiang, we have supplemented the additional interpretations in Sect. 3.2. The impacts of anthropogenic drivers on PM2.5 concentrations in the southern and eastern parts of Zhejiang were evidently weaker than those in other regions in the YRD. This divergence can mostly be explained by spatial distributions of anthropogenic emissions. Anthropogenic emissions in the southern and eastern parts of Zhejiang were also significantly less than those in other regions (Figure S9), thus leading to substantially low PM2.5 concentrations (Figure 3). Besides, meteorological fields in the coastal
- 115 regions, more conducive to PM2.5 diffusions (Figure 5), might be another cause. Added/rewritten part in Sect. 2.4: For all experiments, the prior anthropogenic emissions were kept consistent (i.e., MEIC), while the ECMWF reanalysis datasets accounted for the hourly observational constraints on spatiotemporal meteorological evolutions. The ECMWF reanalysis datasets accounted for the hourly observational constraints on spatiotemporal meteorological meteorological evolutions. Therein almost all necessary meteorological factors (nine variables), involving temperature, U wind
- 120 component, V wind component, pressure, relative humidity, precipitation, short-wave radiation, cloud cover, and planetary boundary layer height (PBLH), were assimilated (https://apps.ecmwf.int/datasets/data/interim-full-daily/levtype=sfc/, last access: 7 March 2020).

Added/rewritten part in Sect. 3.1: In addition, given the fact that the assimilated ERA reanalysis dataset has much wider spatial coverage than ground-based measurements, we also reproduced the spatiotemporal variations in the meteorological

125 factors (e.g., temperature, relative humidity, wind speed, and air pressure) (Figures S5 ~ S8). With the comprehensive evaluation statistics as summarized in Tables S1 ~ S5, it has been demonstrated that the DA method can enable one to derive not only reliable  $PM_{2.5}$  evolutions but also accurate meteorological fields.

Added/rewritten part in Sect. 3.2: We recognized that the impacts of anthropogenic drivers on PM2.5 concentrations in the southern and eastern parts of Zhejiang were evidently weaker than those in other regions in the YRD. This divergence can

130 mostly be explained by spatial distributions of anthropogenic emissions. Anthropogenic emissions in the southern and eastern

of Zhejiang were also significantly less than those in other regions (Figure S9), thus leading to substantially low  $PM_{2.5}$  concentrations (Figure 3). Besides, meteorological fields in the coastal regions, more conducive to  $PM_{2.5}$  diffusions (Figure 5), might be another cause.

4. How did the author consider the regional transport of PM2.5 in this study? The regional emission control effect on PM2.5 may have influence on calculated net impact of emission reduction in each city and the localized mitigation potential.
 Response: Thanks. We agree with the reviewer that regional transport of PM2.5 is central to our results and thus have

considered it carefully. Using observational constraints on the state-of-the-art model, we have reproduced spatiotemporal variations in both PM2.5 and meteorological factors, as illustrated in Sect. 3.1, and thus derived the reliable estimations of
 regional transport of PM2.5. Hence, we have supplemented a sentence in Sect. 3.1 to highlight this point.

- Considering the main objective of this work, we have not conducted source apportionments to predict the impacts of regional transport of  $PM_{2.5}$ . In theory, regional transport of  $PM_{2.5}$  can be attributable to both anthropogenic and meteorological drivers. In turn, we provide paired experiment designs to isolate anthropogenic impacts by subtracting meteorological perturbations (i.e., the differences in simulated  $PM_{2.5}$  concentrations between NO\_2016 and NO\_2019 and between DA\_CON\_G20 and
- 145 NO\_CON\_G20) from the constrained PM2.5 fields (i.e., DA\_2016 and DA\_2019 / DA\_G20).
   Added/rewritten part in Sect. 3.1: Regional transport of PM2.5 can thus be captured reasonably in this way.

[revised manuscript text omitted]

---

## Author Response (AR2)

Nominated Referee #1

**Suggestion comments:**

The authors have addressed most of my comments.

I suggest, however, the abstract still needs a little bit clarification. The following statement is confusing, as it is not immediately clear whether the numbers reported are for long-term or emergency mitigation: "The most substantial declines in PM$_{2.5}$ concentrations ($\sim$ 35 µg/m$^3$, $\sim$ 59%) are achieved in Hangzhou and its surrounding areas. The following hotspots also emerge in megacities, especially in Shanghai (32 µg/m$^3$, 51%), Nanjing (27µg/m$^3$, 55%), and Hefei (24 µg/m$^3$ , 44%)." The previous version was actually better which starts the sentence with "For the winter time periods from 2016 to 2019,..."

**Response:** We thank the reviewer very much and have revised this sentence in Abstract accordingly.

**Added/rewritten part in Abstract:** These emergency measures lead to the largest decrease ( $\sim$ 35 µg/m$^3$, $\sim$ 59%) in PM$_{2.5}$ concentrations in Hangzhou. The hotspots also emerge in megacities, especially in Shanghai (32 µg/m$^3$, 51%), Nanjing (27 µg/m$^3$, 55%), and Hefei (24 µg/m$^3$, 44%) because of the emergency measures.